# Integrative network pharmacology and machine learning identify potential targets of indole-3-lactic acid in colorectal cancer

Jie Li[1], Jian Zhang[1], Jun Ke[1], Zhijian Ren[1]*, Cuncheng Feng[2]*

1 Department of General Surgery, Xi'an International Medical Center Hospital, Xi'an, Shaanxi, China,
2 Department of Gastrointestinal Surgery, The Second People's Hospital of Changzhou, The Third Affiliated Hospital of Nanjing Medical University, Jiangsu, China

* renzhijian126@163.com (ZR); fengcuncheng0612@163.com (CF)

## Abstract

The treatment of colorectal cancer (CRC) remains challenging due to chemotherapy resistance and genetic heterogeneity. Indole-3-lactic acid (ILA), a tryptophan metabolite derived from gut microbiota, exhibits promising anti-inflammatory and anticancer properties; however, its specific molecular targets and regulatory mechanisms in CRC remain poorly understood. In this study, we combined network pharmacology and machine learning with molecular docking to identify candidate targets and pathways for ILA in CRC. We identified 39 ILA-CRC common targets, ultimately identifying four hub genes through the intersection of machine learning models. Validation in independent GEO datasets confirmed significant differential expression of these genes in CRC tissues. Functional enrichment analyses linked these genes to the PPAR, PI3K-AKT, and IL-17 signaling pathways, and gene set enrichment analysis further implicated ascorbate and aldarate metabolism, DNA replication, and fatty acid metabolism. Immune infiltration analysis indicated associations between hub gene expression and immune cell populations, including mast cells, neutrophils, and macrophages, suggesting potential involvement in the tumor immune microenvironment. Molecular docking supported favorable binding of ILA to all four hub proteins, and 100-ns molecular dynamics simulations specifically validated the dynamic stability of the ILA-HMOX1 complex. In conclusion, these results highlight EPHA2, HMOX1, MMP3, and PARP1 as candidate targets and suggest that ILA may influence CRC-related signaling, metabolic programs, and immune contexture, providing a theoretical foundation for developing gut microbiota-derived metabolites as novel anticancer strategies.

## 1. Introduction

Colorectal cancer (CRC), ranking as the third most common malignancy globally and the second leading cause of cancer-related mortality, represents a significant global

**Data availability statement:** Some relevant data are within the manuscript and its Supporting Information files. This research also involves data from the Gene Expression Omnibus database (GEO; http://www.ncbi.nih.gov/geo/), which belongs to the public domain. The accession numbers for the datasets used in this study are GSE44076, GSE74602, GSE32323, and GSE113513.

**Funding:** The research was supported by the Shaanxi Province Natural Science Basic Research Program 2025 (2025JC-YBMS-1051) and the Xi'an International Medical Center Hospital Research Fund (2025QN10).

**Competing interests:** The authors have declared that no competing interests exist.

public health burden. Its pathogenesis is closely associated with chronic inflammation, gut microbiota dysbiosis, and metabolic alterations [1]. While diagnostic and therapeutic advances have improved early-stage outcomes, patients with advanced CRC continue to face dismal prognoses due to persistent challenges, including chemotherapy resistance and treatment-related toxicities [2]. Additionally, the complexity of CRC pathogenesis involves not only genetic heterogeneity but also profound metabolic reprogramming and an immunosuppressive tumor microenvironment (TME). A major bottleneck in current CRC treatment is the limited efficacy of immunotherapies in the majority of patients, largely due to TME-mediated immune evasion [3,4]. Therefore, identifying agents that can simultaneously modulate metabolic programs and reshape the immune contexture remains a critical focus in CRC research.

In the search for new therapeutic avenues, the gut microbiota and its metabolic products have emerged as a critical frontier. During the metabolism of dietary components, microbes generate diverse bioactive metabolites, including short-chain fatty acids and indole derivatives, which can signal through host receptors and shape intestinal physiology [5]. Microbiota dysbiosis may alter both the composition and function of these metabolites, thereby disrupting host immune homeostasis and being associated with tumor initiation and progression [6]. Emerging evidence indicates that microbial metabolites act as direct chemical messengers at the host-microbiota interface. These small molecules can traverse the intestinal barrier and influence key host processes, including immune homeostasis, epigenetic regulation, and metabolic reprogramming [7]. These observations have prompted increasing interest in gut microbiota-derived small molecules as candidates for CRC prevention and adjunctive treatment.

Indole-3-lactic acid (ILA) is a tryptophan-derived microbial metabolite that has been reported to exert anti-inflammatory and antioxidant activities [8]. Recent clinical evidence indicates that fecal ILA levels are significantly reduced in patients with CRC and are inversely associated with tumor stage, suggesting a potential tumor-suppressive role [9]. Mechanistic studies further show that, in colitis-associated colorectal cancer models, ILA can activate aryl hydrocarbon receptor (AhR) signaling in macrophages, influence macrophage differentiation, and attenuate intestinal inflammation, thereby restraining inflammation-driven tumorigenesis [10]. In addition, ILA has been suggested to reshape the tumor immune microenvironment by suppressing M2 macrophage polarization, enhancing CD8+ T-cell cytotoxicity, preserving epithelial barrier integrity, and alleviating inflammation-induced mucosal injury [11,12]. Despite these encouraging findings, the multi-target regulatory network through which ILA interfaces with the genetic complexity and immune contexture of CRC remains incompletely defined and warrants systematic investigation.

In recent years, the rapid development of computational biology approaches, including network pharmacology, machine learning, and molecular docking, has provided powerful tools for elucidating drug mechanisms and screening potential targets. Network pharmacology systematically reveals intricate relationships between drugs, targets, and diseases, through multi-source data integration, enabling researchers to comprehensively understand pharmacological mechanisms from a holistic

perspective [12]. Machine learning algorithms effectively identify critical genes and potential targets from large-scale datasets, markedly enhancing drug screening efficiency and accuracy [13]. Molecular docking technology validates binding affinity and stability by simulating ligand-protein interactions, thereby providing structural insights for rational drug design. Integrating these methodologies accelerates drug discovery processes and establishes a robust theoretical foundation for clinical applications. In this study, we integrated machine learning with network pharmacology to identify potential targets and elucidate the molecular mechanisms of ILA in CRC. Subsequent molecular docking and molecular dynamics simulations validated these computational predictions. Our findings provide novel theoretical foundations and promising putative targets for applying gut microbiota-derived metabolites in CRC therapy. Fig 1 shows a schematic workflow for this study.

## 2. Materials and methods

### 2.1. Collection of ILA targets

The PubChem database (https://pubchem.ncbi.nlm.nih.gov/) (accessed on 8 January 2025) [14] was used to obtain molecular structural formulae and canonical SMILES information for ILA. Subsequently, five online public databases were employed to predict ILA-related targets: SwissTargetPrediction (http://www.swisstargetprediction.ch/) (accessed on 8 January 2025) [15], Similarity Ensemble Approach (https://sea.bkslab.org/) (accessed on 8 January 2025) [16], TargetNet (http://targetnet.scbdd.com/home/index/) (accessed on 8 January 2025) [17], SuperPred (http://prediction.charite.de/) (accessed on 8 January 2025) [18], and PharmMapper (http://www.lilab-ecust.cn/pharmmapper) (accessed on 8 January 2025) [19].

For SwissTargetPrediction, SEA, SuperPred, and PharmMapper, all predicted targets annotated as Homo sapiens were retained without applying additional score or probability cutoffs, as these platforms primarily provide ranked or similarity-based predictions rather than unified probability thresholds. This strategy was adopted to ensure comprehensive coverage of potential human targets at the initial screening stage. For TargetNet, which provides explicit prediction probabilities, only targets with a probability greater than 0 were included according to the database output criteria. Predicted targets obtained from the five databases were subsequently merged, and duplicate entries were removed. All target proteins were standardized to human species genes via the UniProt database (https://www.uniprot.org/) (accessed on 9 January 2025) [20]. Finally, the ILA target network was created using Cytoscape 3.9.1 (https://cytoscape.org/) (accessed on 9 January 2025).

### 2.2. Collection of CRC targets

We conducted a comprehensive search using the keywords "colorectal cancer" and "colorectal carcinoma" to identify CRC-related targets from four databases: OMIM (http://www.omim.org) (accessed on 10 January 2025) [21], TTD (https://db.idrblab.org/ttd/) (accessed on 10 January 2025) [22], GeneCards (https://www.genecards.org/) (accessed on 10 January 2025) [23], and DrugBank (https://go.drugbank.com/) (accessed on 10 January 2025) [24]. For GeneCards, we applied a relevance score cutoff set at or above the median value. The targets from the four databases were combined, and duplicates were deleted. At the same time, we retrieved the microarray dataset from the Gene Expression Omnibus database (GEO; http://www.ncbi.nih.gov/geo/) (accessed on 10 January 2025) [25], and the datasets were filtered based on the following inclusion criteria: (1) organism restricted to Homo sapiens; (2) study design comparing primary colorectal cancer tissues vs. normal colonic tissues; (3) gene expression profiling by array; and (4) sufficient sample size (>25 samples) to ensure statistical reliability. Based on these criteria, four publicly available CRC-related microarray datasets (GSE44076, GSE74602, GSE32323, and GSE113513) were selected for subsequent analyses. These datasets were chosen because they provide high-quality transcriptomic profiles with sufficient sample sizes and have been widely used or are representative datasets in CRC-related gene expression studies. Among them, GSE44076 was designated as the training dataset, while GSE74602, GSE32323, and GSE113513 were used as external validation datasets. Expression matrices from the validation datasets were extracted using Perl software (version 5.30.2) and subsequently integrated.

 

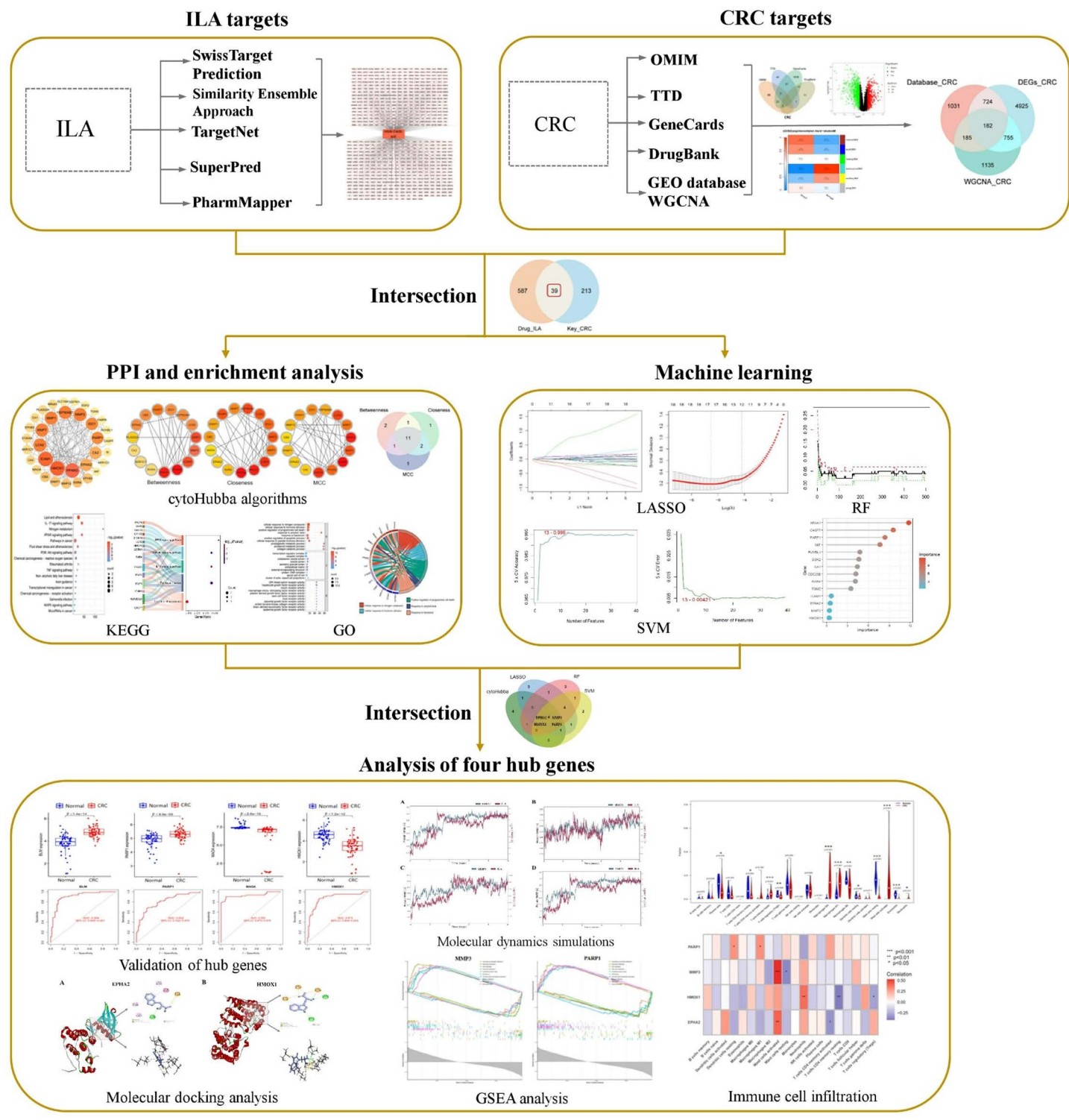

**Fig 1. Workflow chart.**

Batch effects arising from differences among datasets were corrected using the limma (version 3.64.1) and sva (version 3.56.0) packages implemented in R software (version 4.4.2), resulting in normalized gene expression data suitable for downstream bioinformatic analyses. Detailed information for all included datasets is summarized in Table 1.

## 2.3. Identification of differentially expressed genes (DEGs)

Differential expression analysis was performed between normal and CRC groups using the limma package in R, with significance thresholds set at $|logFC| > 1$ and adjusted p-value $< 0.05$. The resulting DEGs were then visualized through a volcano plot and heatmap, generated using the ggplot2 (version 3.5.2) and pheatmap (version 1.0.12) packages, respectively.

## 2.4. Weighted gene co-expression network analysis

Weighted gene co-expression network analysis (WGCNA) is a systems biology method that enables the exploration of high-dimensional gene expression datasets through network-based approaches, aiming to confirm gene modules and key genes associated with biological processes to reveal potential biological pathways or mechanisms [26]. First, we preprocessed the raw gene expression data and constructed a gene relationship matrix. We determined the optimal soft threshold power (β) using the pickSoftThreshold function in the WGCNA R package. The criterion for selection was to achieve a scale-free topology fit index ($R^2$) of at least 0.85 while maintaining a reasonable mean connectivity. Consequently, a power of β = X was selected based on the scale independence and mean connectivity plots. Using the selected β value, an adjacency matrix was constructed and transformed into a topological overlap matrix (TOM), which reflects the network connectivity between gene pairs. The dissimilarity measure (1-TOM) was then used as the distance metric for hierarchical clustering to identify gene modules. Modules were detected using the dynamic tree cut algorithm with a predefined minimum module size, and highly similar modules were merged based on module eigengene correlation to improve module robustness. To assess the association between gene modules and CRC-related traits, module-trait relationships were evaluated by calculating Pearson correlation coefficients between module eigengenes and phenotypic traits. Gene significance (GS) and module membership (MM) values were further computed to identify biologically relevant core modules and genes most strongly associated with CRC for downstream analyses.

## 2.5. Construction of protein-protein interaction (PPI) network

We identified potential key genes for CRC by the intersection of databases, DGEs, and core module genes. The overlapping targets between ILA and CRC were identified using Venn diagram analysis (http://www.bioinformatics.com.cn/). These common targets were subsequently analyzed in the STRING database (version 12.0; https://string-db.org) [27] with the following parameters: organism restricted to Homo sapiens and minimum interaction confidence score set to >0.4 (medium confidence). The PPI data were downloaded from the STRING database in TSV format and imported into Cytoscape software (version 3.9.1) for network construction and visualization. To analyze the topological characteristics, the 'Analyze Network' tool was utilized with the interaction type set to 'Treat network as undirected'. Node centrality measures were then calculated using the CytoNCA plugin (version 2.1.6), with degree centrality used as the primary metric. Isolated

**Table 1. Detailed information for the GEO datasets.**

| Dataset | GEO accession | Year | Platform | Disease | Control |
|---|---|---|---|---|---|
| Training set | GSE44076 | 2014 | GPL13667 | 98 | 50 |
| Validation set | GSE74602 | 2016 | GPL6104 | 30 | 30 |
| | GSE32323 | 2012 | GPL570 | 17 | 17 |
| | GSE113513 | 2018 | GPL15207 | 14 | 14 |

nodes without interactions (degree = 0) were excluded from the network. Finally, to intuitively visualize the network structure, the 'Style' panel was used to map node attributes: node size and color were adjusted according to their DC values, where larger and darker nodes represented higher connectivity. We employed three topological algorithms from the cytoHubba plugin (version 0.1)- betweenness, closeness, and maximal clique centrality (MCC) – to identify hub genes within the PPI network.

## 2.6. Enrichment analysis

We employed the Metascape online tool (version 3.5; http://metascape.org/) [28] to perform Gene Ontology (GO) annotations and Kyoto Encyclopedia of Genes and Genomes (KEGG) pathway enrichment analysis of the selected genes. The GO enrichment analysis included the three domains: biological process (BP), cellular component (CC), and molecular function (MF). The resulting data were visualized in an online visualization platform (http://www.bioinformatics.com.cn) [29]. Additionally, we performed gene set enrichment analysis (GSEA) on hub genes using the clusterProfiler (version 4.16.0) R package. The enriched biological pathways relevant to the predicted mechanisms of ILA in CRC were subsequently visualized using the enrichplot (version 1.28.4) and pathview (version 1.30) packages.

## 2.7. Determination of hub genes with machine learning algorithms

In this study, three machine learning algorithms, least absolute shrinkage and selection operator (LASSO), random forest (RF), and support vector machine recursive feature elimination (SVM-RFE), were applied to jointly identify hub genes associated with the hypothesized regulatory roles of ILA in CRC. To ensure reproducible results, we set the random seed to 12345 [30]. To address the sample imbalance between CRC and normal groups, stratified cross-validation was employed across all models to maintain consistent class proportions. Specifically, LASSO employs L1 regularization to mitigate multicollinearity and induce sparsity within high-dimensional datasets; RF was selected to capture complex non-linear interactions; and SVM-RFE identifies the optimal feature subset to maximize classification performance. These three algorithms complement each other, and their integration minimizes the bias inherent in any single model, thereby enhancing the robustness of the screening results.

The LASSO regression analysis was performed using the glmnet package (version 4.1.10) [31]. Specifically, parameters included family = binomial (binary classification) and alpha = 1 (LASSO penalty) to mitigate overfitting. To optimize hyperparameters, a 10-fold cross-validation procedure was performed using deviance as the optimization metric. The optimal lambda value (lambda.min) was selected based on the minimum mean cross-validated deviance. Features with non-zero coefficients at this optimal lambda value were retained as significant targets.

RF modeling was implemented using the randomForest package (version 4.7.1.2) [32]. The number of decision trees was evaluated across a range from 1 to 500, and the optimal tree number was determined based on the out-of-bag (OOB) error rate to balance model performance and complexity. An optimized RF model was then constructed using this tree number. Feature importance scores were calculated using the mean decrease in accuracy metric. Genes with importance scores greater than 0.3 were retained as candidate features for downstream analysis. This threshold was applied as a pragmatic filtering criterion to retain relatively high-importance features while reducing noise prior to integration with other algorithms. Sensitivity analyses using alternative thresholds confirmed that the final consensus hub genes were robust to reasonable variations in this cutoff.

SVM-RFE was performed using the svmRadial, e1071 (version 1.7.16), and caret (version 7.0.1) packages [33]. Before SVM training, all features were standardized using z-score normalization to ensure comparable feature scales. To avoid data leakage, normalization parameters were estimated exclusively on the training folds and subsequently applied to the corresponding validation folds. A five-fold stratified cross-validation scheme was employed during recursive feature elimination. The optimal gene subset was selected based on the lowest mean classification error and the highest cross-validated predictive performance.

Finally, genes consistently identified by all three algorithms (LASSO, RF, and SVM-RFE) were defined as high-confidence hub genes associated with the predicted regulatory mechanisms of ILA in CRC.

## 2.8. Validation of hub genes

In this study, GSE44076 was used as a training set containing 148 samples (50 normal and 98 CRC samples). Three independent datasets (GSE74602, GSE32323, and GSE113513) were integrated to form a validation set comprising 122 samples (61 normal and 61 CRC samples). Differential expression patterns of hub genes and their diagnostic performance were assessed using the limma (version 3.64.1), ggpubr (version 0.6.0), and pROC (version 1.18.5) packages in R, which generated comparative box plots and receiver operating characteristic (ROC) curves for both sample groups.

## 2.9. Immune cell infiltration

We performed immune cell infiltration analyses using the CIBERSORT algorithm [34], which employs linear support vector regression to quantify the relative proportions of 22 distinct immune cell subtypes in both CRC patients and normal subjects. Following this analysis, we identified immune cell populations demonstrating statistically significant differences between the two groups and evaluated their correlations with hub gene expression levels using the Spearman method.

## 2.10. Molecular docking analysis

Protein and ligand structures were prepared using the Protein Preparation Wizard and LigPrep modules in the Schrödinger Suite 2021−4, with energy minimization performed under the OPLS4 force field [35]. The crystal structures of EPHA2, HMOX1, MMP3, and PARP1 were obtained from the Protein Data Bank (PDB; https://www.rcsb.org/structure) and processed by assigning bond orders, adding hydrogens, optimizing hydrogen-bond networks, and minimizing heavy atoms to relieve steric clashes. The ligand (ILA) was prepared with LigPrep by generating possible ionization/tautomeric states at physiological pH and by enumerating low-energy conformers. The receptor grids were generated using the Receptor Grid Generation panel in Glide. The center of the grid box was defined by the centroid of the co-crystallized ligand within the active site of each target protein. The bounding box was sized to sufficiently enclose the active site and accommodate the ligand. Docking was performed using Glide Standard Precision (SP) with flexible ligand sampling enabled, including ring conformation sampling, and post-docking minimization was applied. Van der Waals radii scaling was set to 0.8 for ligand atoms with a partial charge cutoff of 0.15 (receptor scaling: 1.0, charge cutoff 0.25) to allow for limited softness in the potential. The resulting poses were subjected to post-docking minimization, and the best-scored pose (lowest Glide score) was selected for further analysis.

## 2.11. Molecular dynamics (MD) simulations

The highest-ranked docking poses of the four compounds were used as initial structures for MD simulations, performed with Desmond in the Schrödinger Suite 2021−4 [35]. Each complex was solvated in an orthorhombic TIP3P water box and neutralized with counterions; NaCl was added to a final concentration of 0.15 M. Simulations were conducted using the OPLS4 force field. Systems were first subjected to energy minimization followed by equilibration using the standard Desmond relaxation protocol, after which production runs were carried out under the NPT ensemble at 300 K and 1 atm. Temperature and pressure were controlled using the default Desmond thermostat and barostat settings for Schrödinger Suite 2021−4. For comparative assessment, each of the four complexes (ILA bound to EPHA2, HMOX1, MMP3, and PARP1) was simulated for 10 ns under identical conditions. Based on the comparative stability, an additional 100 ns production simulation was performed for the ILA-HMOX1 complex. Trajectory analyses included root mean square deviation (RMSD) to evaluate global conformational stability, root mean square fluctuation (RMSF) of protein Cα atoms to assess residue-level flexibility, and protein-ligand interaction analyses to characterize the persistence and types of intermolecular

contacts over time. The binding free energy values and interactions of ligands with proteins were calculated by the MM/GBSA method.

## 3. Results

### 3.1. Acquisition of the compound ILA targets

The chemical structure of ILA is presented in Fig 2A. We retrieved 100, 150, 140, 92, and 297 ILA-related targets from five network pharmacology databases, respectively (S1 File in S1 Data). After removing duplicate entries, 626 unique putative ILA targets were retained. The overlap of ILA-related targets identified across the five databases is summarized in the Venn diagram (Fig 2B). These targets were subsequently visualized as an ILA-target interaction network constructed in Cytoscape (Fig 2C).

### 3.2. Acquisition of disease CRC targets

From four databases, we obtained 93, 95, 2005, and 25 CRC-related targets, respectively. After removing duplicates, 2122 unique CRC-related targets remained (Fig 3A; S2 File in S1 Data). Using the CRC-related GEO dataset GSE44076 as the training set, we identified 1891 DEGs, comprising 891 upregulated and 1000 downregulated genes. The DEG distribution is shown in the volcano plot (Fig 3B), and the expression patterns of the top 50 most significant DEGs are

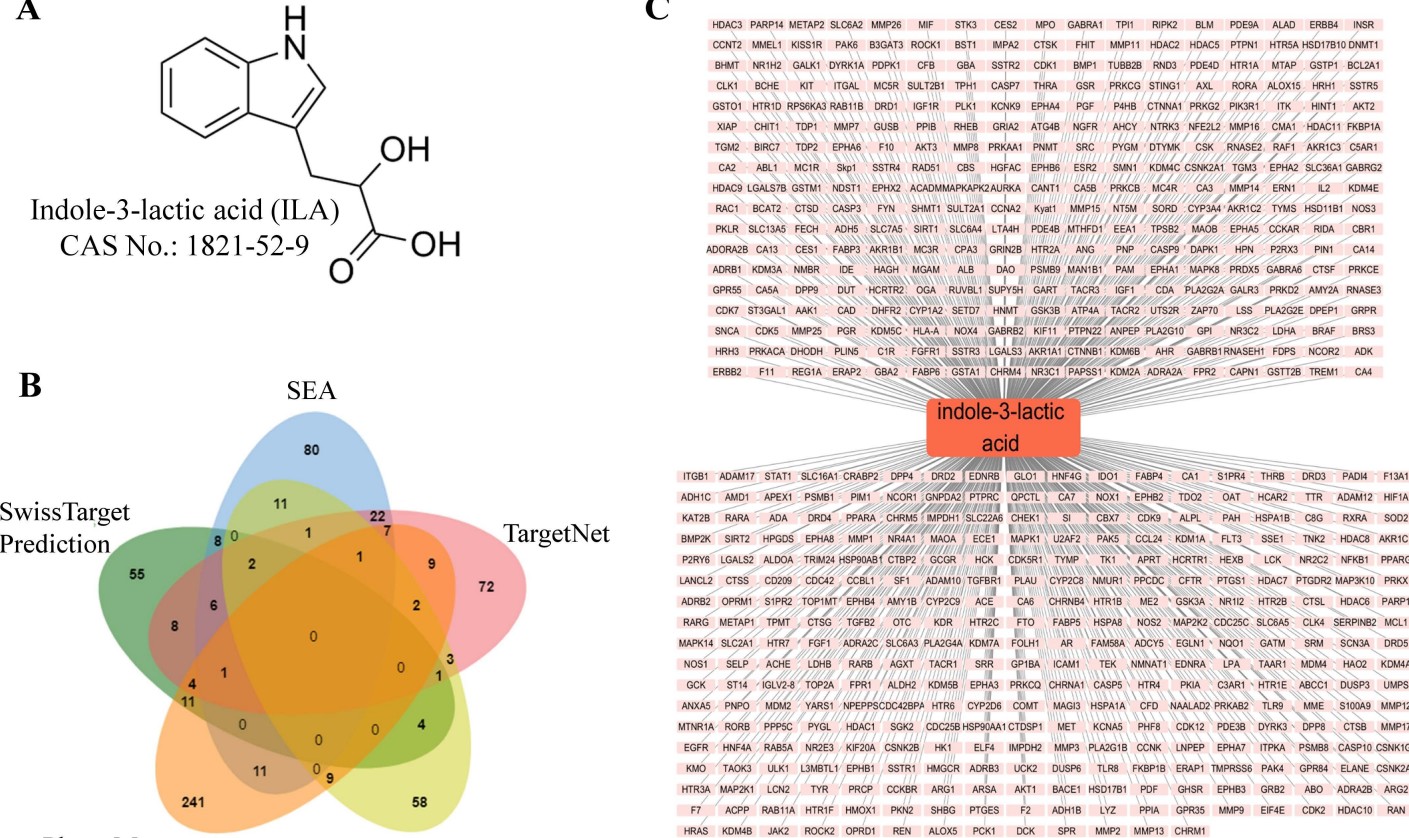

**Fig 2. Identification of ILA targets via network pharmacology.** (A) The chemical structure of ILA. (B) Venn diagram showing ILA-related targets among the five databases. (C) ILA-targets interaction network.

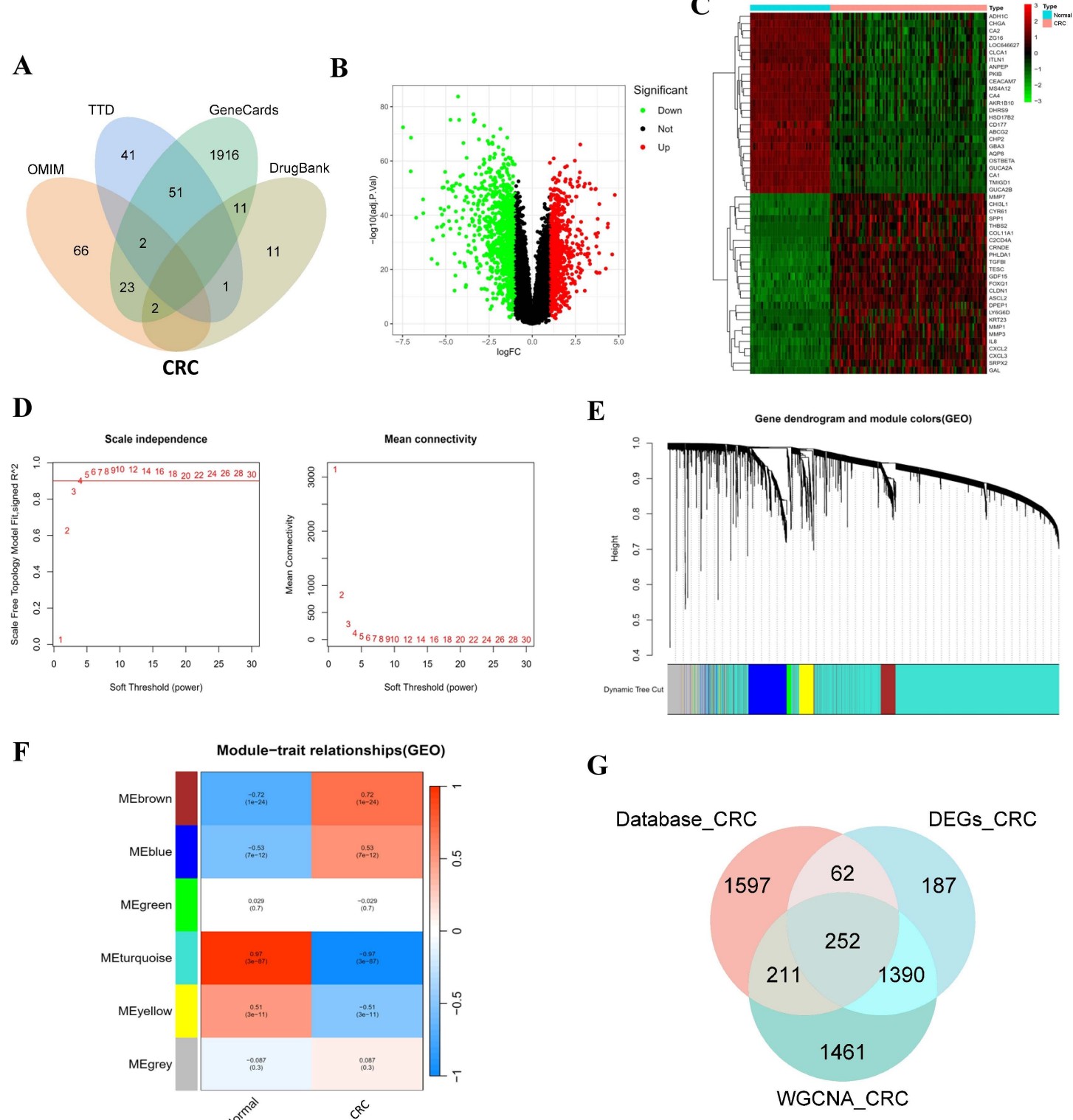

**Fig 3. Integrated identification of CRC targets through multi-database mining and WGCNA. (A)** Venn diagram of CRC targets retrieved from four databases. **(B)** Volcano plot of DEGs in CRC vs. normal groups. Red points: 891 significantly upregulated genes; green points: 1000 downregulated

genes. **(C)** Hierarchical clustering heatmap of the top 50 most significant DEGs. **(D)** Scale independence and average connectivity to determine soft thresholds in WGCNA. **(E)** Gene clustering dendrogram and highly correlated gene modules. **(F)** The heatmap of module–trait relationships for the 6 co-expression modules. MEturquoise module showed the strongest positive correlation with CRC (R = 0.97, p < 0.0001). **(G)** Intersection of CRC targets from databases (2122 genes), DEGs (1891 genes), and WGCNA (3314 genes), identifying 252 CRC key genes.

displayed in the hierarchical clustering heatmap (Fig 3C), indicating clear separation between normal and CRC samples. The validation set was obtained by merging the three data sets GSE74602, GSE32323, and GSE113513. To evaluate the effectiveness of data integration, we performed Principal Component Analysis (PCA). As shown in S1 Fig in S1 Data, the samples from different datasets were clearly separated before correction, indicating a significant batch effect. However, after batch effect correction, the samples from the three datasets were well-mixed and uniformly distributed, demonstrating that the non-biological variations were successfully removed for subsequent analyses. To identify CRC-related gene modules, we constructed a weighted gene co-expression network using WGCNA. Based on scale independence and average connectivity, the optimal soft threshold was set at 4, achieving a scale-free index of 0.9 with favorable mean connectivity (Fig 3D). The resulting gene clustering dendrogram (Fig 3E) identified distinct co-expression modules represented by different-colored branches. After merging similar modules, six distinct gene modules were obtained. The MEturquoise module showed the strongest association with normal/CRC phenotypes (R = 0.97, P < 0.0001; Fig 3F) and contained 3314 genes prioritized as potential CRC targets. Finally, intersecting CRC-related genes from databases (2122 genes), DEGs (1891 genes), and MEturquoise-module genes (3314 genes) yielded 252 CRC key genes (Fig 3G).

### 3.3. PPI network and enrichment analysis of common targets associated with ILA in CRC

Based on the intersection analysis above, 39 potential targets of ILA against CRC were identified (Fig 4A). A PPI network of these 39 targets was constructed using STRING to characterize target-target interactions (Fig 4B). We then applied the cytoHubba plugin to rank hub candidates; the top 15 targets based on betweenness, closeness, and MCC are shown in Fig 4C, and their intersection yielded 11 core genes (Fig 4D): ICAM1, MMP3, HMOX1, LCN2, PPARG, MMP7, IDO1, PARP1, HSP90AB1, EPHA2, and CBS. KEGG and GO enrichment analyses based on these 39 targets were performed to identify biological processes and signaling pathways most likely involved in the predicted regulatory roles of ILA in CRC. KEGG pathway analysis revealed 17 significantly enriched pathways (P < 0.05) (S2 Fig in S1 Data), with the top five being lipid and atherosclerosis, pathways in cancer, IL-17 signaling pathway, PI3K-Akt signaling pathway, and PPAR signaling pathway. The core targets associated with these pathways included HSP90AB1, ICAM1, MMP1, MMP3, and HMOX1; the associations between key targets and the top pathways are visualized in the Sankey bubble chart (Fig 4E). GO analysis identified 247 BPs, 15 CCs, and 66 MFs, with the top 10 enriched terms shown in S2 Fig in S1 Data. The chord diagram (Fig 4F) highlights the top five BPs: cellular responses to nitrogen compounds, cellular responses to hormonal stimuli, positive regulation of programmed cell death, response to amyloid-beta, and response to bacterium (S3 File in S1 Data).

### 3.4. Selection of target hub genes via machine learning

To further identify critical hub genes for ILA against CRC, we screened the 39 targets using three machine-learning algorithms. For the LASSO regression, the changing trajectory of independent variable coefficients and the partial likelihood deviance were plotted, identifying 17 core targets at the optimal lambda value (Fig 5A). In the RF algorithm, the error rate stabilized as the number of decision trees increased; we selected 14 genes with a relative importance score greater than 0.3 for downstream analysis (Fig 5B). The SVM-RFE algorithm selected 13 genes with the lowest error (Fig 5C). S4 File in S1 Data presents a ranked list of core genes identified by the three machine learning algorithms. Finally, intersecting the candidates from cytoHubba, LASSO, RF, and SVM-RFE identified four hub genes: EPHA2, HMOX1, MMP3, and PARP1 (Fig 5D).

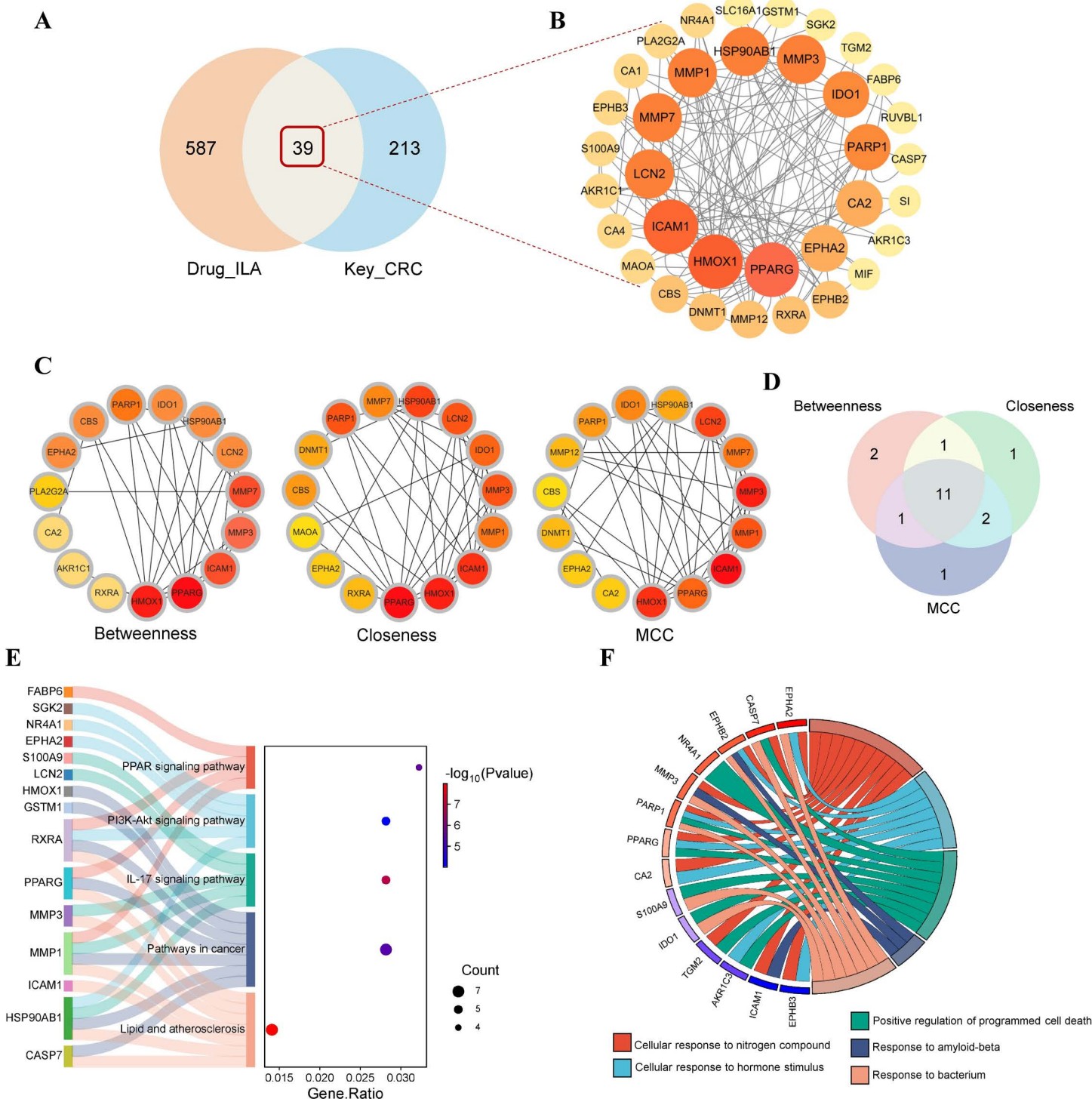

**Fig 4. PPI network and functional enrichment analysis of common targets associated with ILA in CRC. (A)** Venn diagram identifying 39 potential targets between ILA and CRC. **(B)** PPI network of 39 common targets. **(C)** Top 15 hub genes ranked by cytoHubba plugin using three topological algorithms. **(D)** Eleven intersection hub genes of the betweenness, closeness, and MCC algorithms. **(E)** Sankey bubble chart of the top 5 KEGG pathways. **(F)** Chord diagram of the top 5 biological processes.

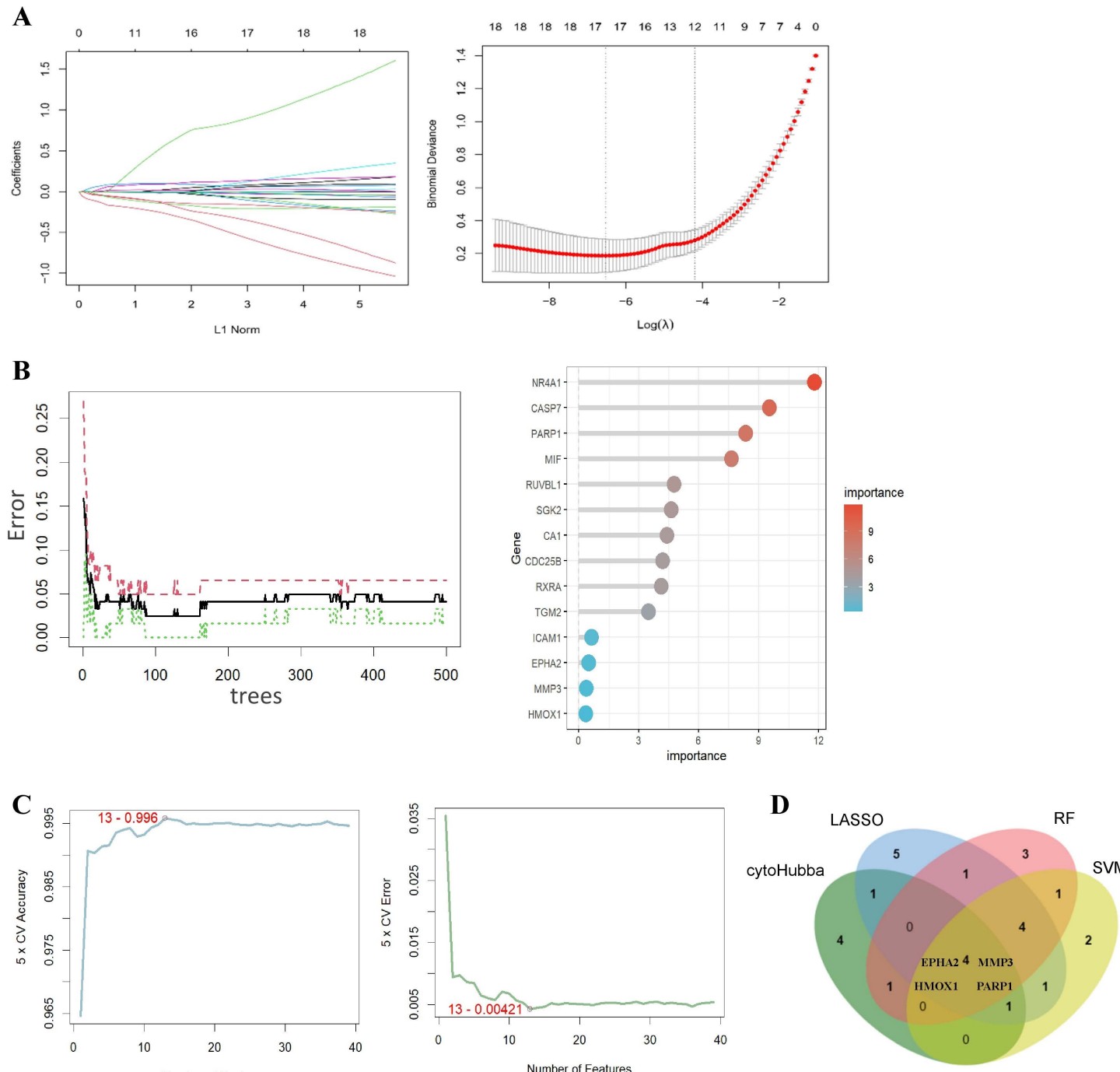

**Fig 5. Machine learning-based identification of candidate targets associated with ILA in CRC. (A)** The coefficients and regularization plot of LASSO regression, the vertical dashed lines indicate the optimal lambda value. **(B)** Error rates in random forests and the top 14 genes with relative importance greater than 0.3. **(C)** Accuracy and error rate curves of SVM-REF algorithm. **(D)** Four hub genes (EPHA2, HMOX1, MMP3, PARP1) were identified by intersecting cytoHubba and machine learning selected targets.

### 3.5. Detection of the expression of hub genes in CRC

We next examined the expression patterns of the four hub genes between the normal and CRC groups. In the GSE44076 training set, EPHA2, MMP3, and PARP1 were significantly upregulated in CRC, whereas HMOX1 was significantly downregulated (Fig 6A). The area under the curve (AUC) values of ROC were 0.939 for EPHA2, 0.980 for HMOX1, 0.955 for MMP3, and 0.997 for PARP1 (Fig 6B). We further validated these findings in an independent merged validation cohort (GSE74602 + GSE32323 + GSE113513). The expression trends remained consistent (Fig 6C), and ROC analysis confirmed the diagnostic value of these hub genes (Fig 6D), with the AUC values of ROC were 0.935 for EPHA2, 0.883 for HMOX1, 0.935 for MMP3, and 0.804 for PARP1. These results support the robustness of the identified hub genes in distinguishing CRC from normal samples across datasets.

### 3.6. GSEA analysis

GSEA analysis identified signaling pathways associated with the four hub genes, with the top 10 upregulated and downregulated pathways shown in Fig 7A-D. EPHA2 was significantly associated with aminoacyl-tRNA biosynthesis, ascorbate and aldarate metabolism, DNA replication, and fatty acid metabolism (Fig 7A). HMOX1 expression was significantly correlated with ascorbate and aldarate metabolism, DNA replication, and fatty acid metabolism (Fig 7B). MMP3 was significantly correlated with ascorbate and aldarate metabolism, butanoate metabolism, and fatty acid metabolism (Fig 7C). PARP1 was significantly associated with ascorbate and aldarate metabolism, butanoate metabolism, and DNA replication (Fig 7D). Overall, these results suggest that the hub genes are closely linked to pathways involving redox-related metabolism, DNA replication, and lipid metabolism.

### 3.7. Immune cell infiltration

Given the close relationship between CRC progression and the tumor immune microenvironment, we assessed the relative abundance of 22 immune cell subtypes using CIBERSORT. Differential immune cell infiltration between normal and CRC samples in the GSE44076 training set is shown in Fig 8A. Compared with the normal group, plasma cells, T cells follicular helper, macrophages M0/M1/M2, activated mast cells, and neutrophils were significantly increased in CRC, whereas resting CD4 memory T cells, Tregs, resting mast cells, and eosinophils were significantly decreased.

We further evaluated the associations between the four hub genes and immune cell infiltration; the Spearman correlation heatmap is provided in Fig 8B. EPHA2 showed a significant positive correlation with activated mast cells and a significant negative correlation with plasma cells. HMOX1 was positively correlated with neutrophils and negatively correlated with resting CD4 memory T cells and Tregs. MMP3 was positively correlated with activated mast cells and negatively correlated with resting mast cells. PARP1 was negatively correlated with resting dendritic cells and macrophages M1. These results indicate that hub gene expression is linked to distinct immune infiltration patterns in CRC.

### 3.8. Molecular docking

Molecular docking was conducted to evaluate the binding affinity and predicted binding poses of ILA with the four hub genes. The docking conformations of ILA with EPHA2, HMOX1, MMP3, and PARP1 are shown in Fig 9A-D, respectively. All four complexes exhibited favorable docking scores (EPHA2: −6.247; HMOX1: −5.876; MMP3: −7.208; PARP1: −5.857), suggesting stable binding potential. In the EPHA2-ILA complex (Fig 9A), ILA formed π-π stacking interactions with TYR694, π-alkyl interactions with LEU746, ALA644, and ILE619, and a hydrogen bond with THR692. The predicted binding poses for HMOX1, MMP3, and PARP1 (Fig 9B-D) further support that ILA can be accommodated within their binding pockets with favorable interaction patterns, consistent with the docking affinities reported in Table 2. Overall, these docking results suggest that ILA may be involved in CRC-associated regulatory processes through interactions with these proteins.

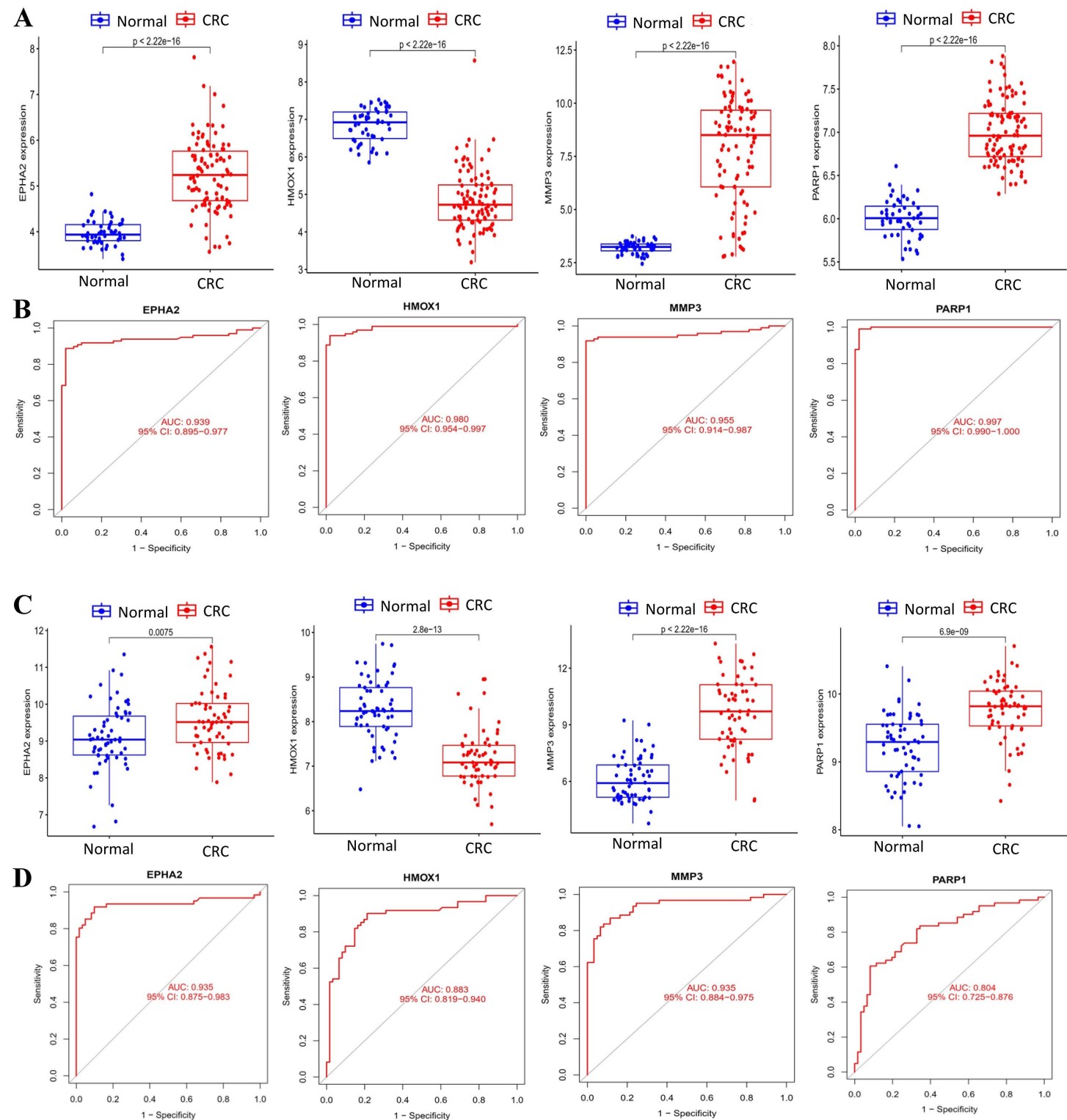

**Fig 6. Expression analysis of hub genes in the normal and CRC groups. (A)** Differential expression of EPHA2, HMOX1, MMP3, and PARP1 between normal (n=50) and CRC (n=98) in the GSE44076 training set. **(B)** ROC curves demonstrating diagnostic performance of hub genes in GSE44076. AUC values: EPHA2 (0.939), HMOX1 (0.980), MMP3 (0.955), PARP1 (0.997). **(C)** Differential expression of hub genes between normal (n=61) and CRC (n=61) in the validation set. **(D)** ROC analysis of hub genes in the validation set. AUC values: EPHA2 (0.935), HMOX1 (0.883), MMP3 (0.935), PARP1 (0.804).

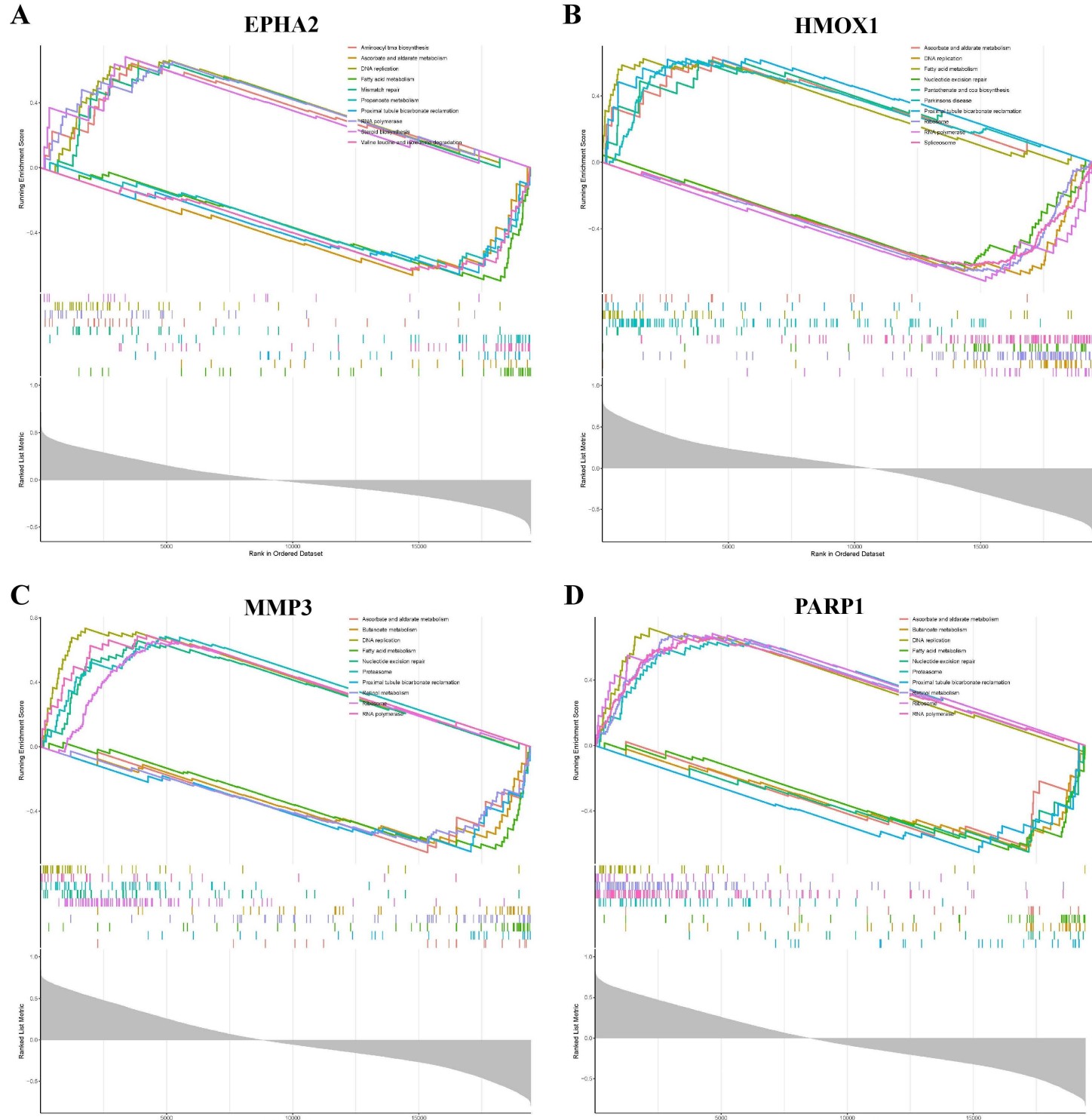

**Fig 7. GSEA analysis of the hub genes. (A)** GSEA up- and down-regulation pathways for EPHA2. **(B)** GSEA up- and down-regulation pathways for HMOX1. **(C)** GSEA up- and down-regulation pathways for MMP3. **(D)** GSEA up- and down-regulation pathways for PARP1.

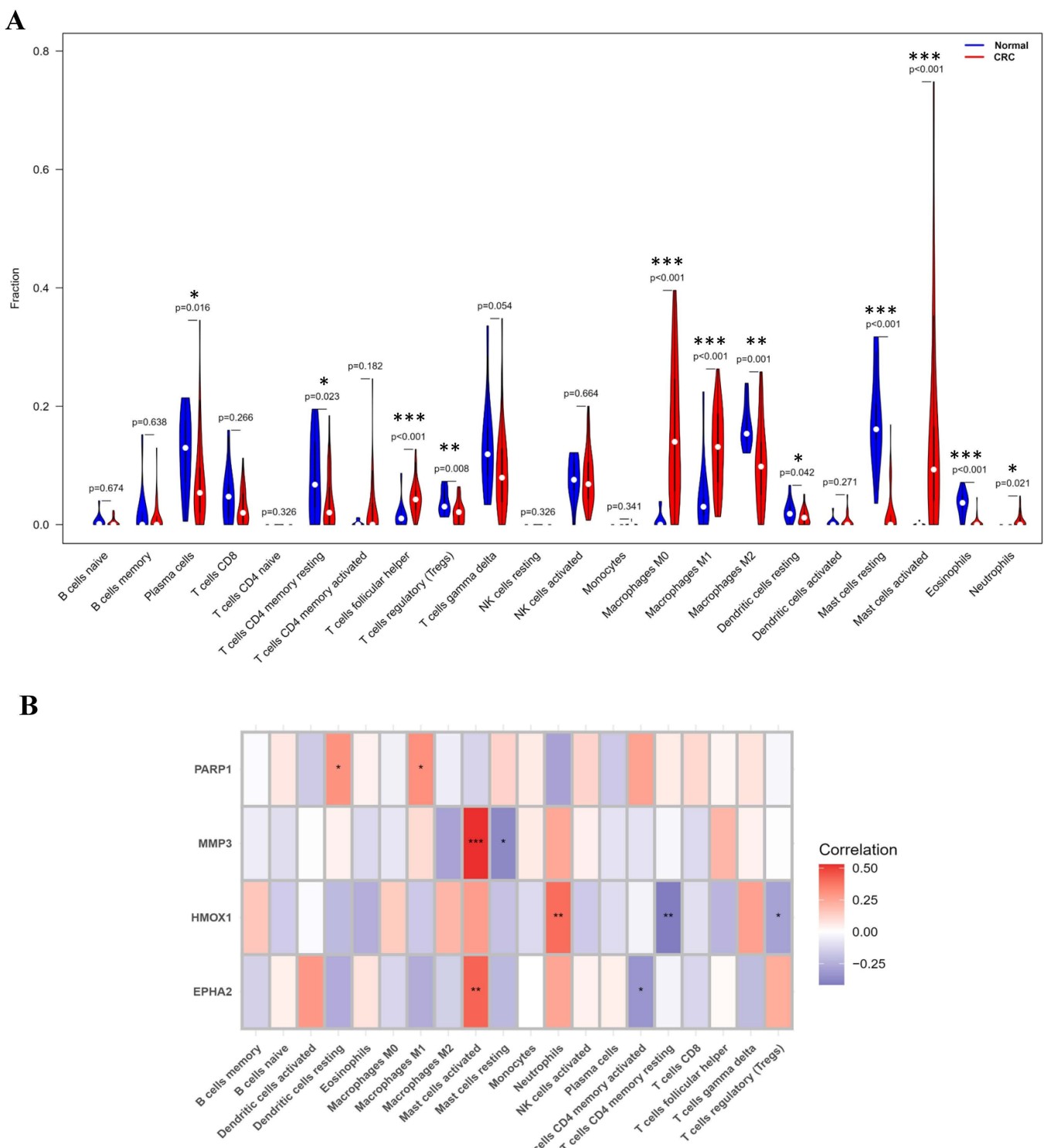

**Fig 8. Immune cell infiltration analysis. (A)** Differential immune cell infiltration between normal (n = 50) and CRC (n = 98) in the GSE44076 training set. CIBERSORT analysis quantified 22 immune cell subtypes. **(B)** Spearman correlation heatmap between hub genes and immune cell subtypes. Asterisks denote statistical significance of correlation coefficients: *P < 0.05, **P < 0.01, ***P < 0.001.

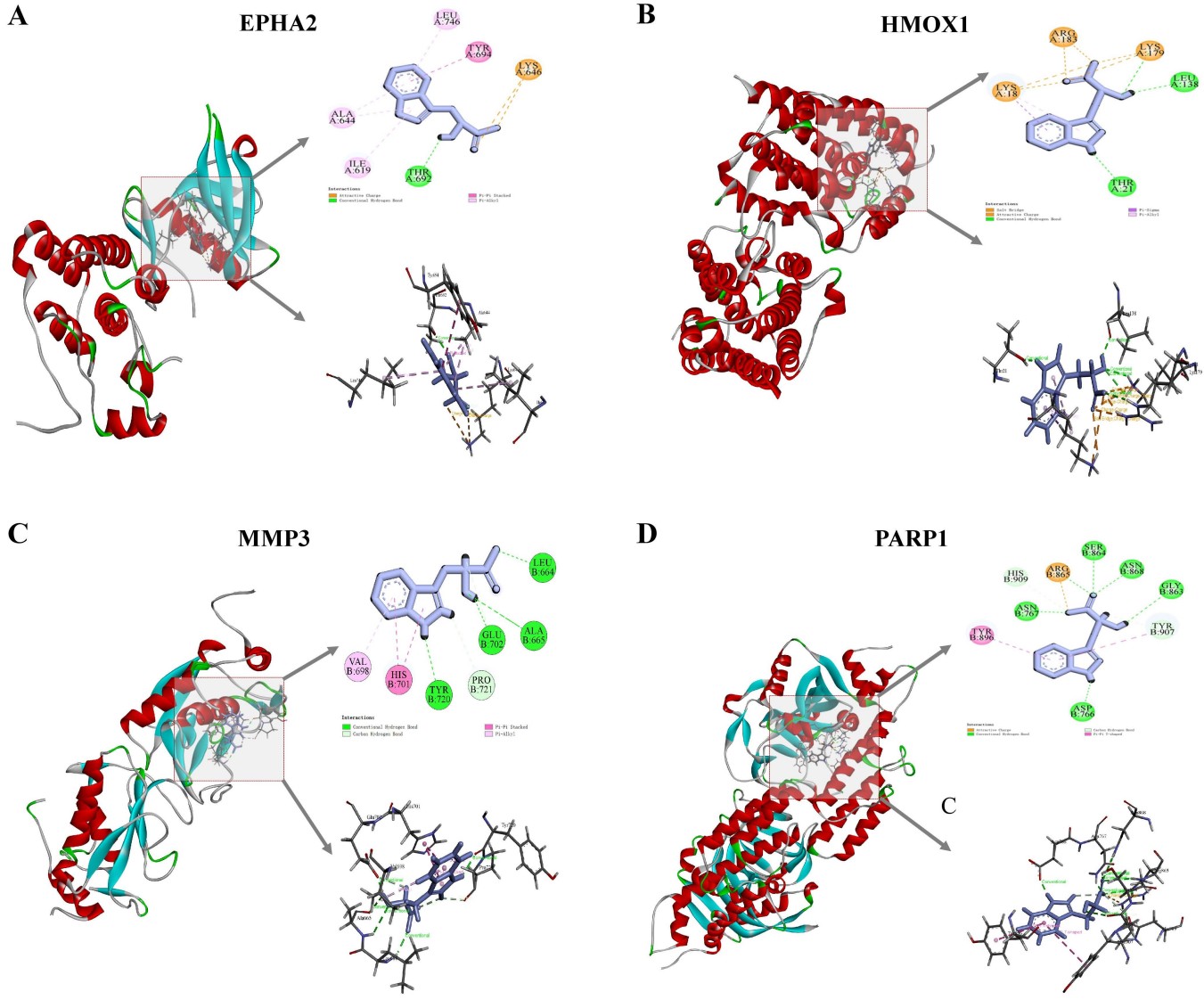

**Fig 9. Molecular docking analysis of ILA and four hub genes. (A)** Docking diagrams for ILA and EPHA2, binding affinity −6.247 kcal/mol. **(B)** Docking diagrams for ILA and HMOX1, binding affinity −5.876 kcal/mol. **(C)** Docking diagrams for ILA and MMP3, binding affinity −7.208 kcal/mol. **(D)** Docking diagrams for ILA and PARP1, binding affinity −5.857 kcal/mol.

**Table 2. Detailed parameters of molecular docking.**

| Number | Genes | PDB ID | Box_center (x, y, z)/Å | Affinity/(kcal/mol) |
|---|---|---|---|---|
| 1 | EPHA2 | 6Q7D | −82.93, −19.24, 89.25 | −6.247 |
| 2 | HMOX1 | 1N45 | 18.41, −0.96, 1.4 | −5.876 |
| 3 | MMP3 | 1HY7 | 2.03, 48.73, 57.68 | −7.208 |
| 4 | PARP1 | 7KK2 | 3.54, −17.64, 31.88 | −5.857 |

### 3.9. Molecular dynamics simulations of ILA with hub genes

To assess the dynamic stability of the docked poses and validate the binding interactions, MD simulations were performed. Comparative RMSD profiles for the four protein-ILA complexes over a 10 ns simulation are shown in Fig 10A-D, indicating that the ILA-HMOX1 complex exhibited the most rapid equilibration and the lowest RMSD fluctuation among the four candidates. It should be acknowledged that the 10 ns MD trajectories represent a preliminary stability assessment and are insufficient to fully characterize long-timescale conformational changes. Therefore, conclusions derived from the short simulations should be interpreted cautiously and were complemented by extended sampling for the representative ILA-HMOX1 complex. The 100 ns simulation confirmed the dynamic stability of the ILA-HMOX1 complex (Fig 10E). The RMSD of the protein Cα atoms converged after approximately 20 ns, maintaining a stable value of approximately 1.8 Å. Critically, the ligand RMSD, when fitted to the protein, also remained stable within a narrow range (fluctuating around 2.2 Å), indicating that ILA maintained a consistent binding pose within the active site without dissociating. Besides, results of MM/GBSA showed that the binding free energy for ILA to HMOX1 protein was −25.58 kcal/mol.

To further evaluate residue-level flexibility, RMSF analysis of protein Cα atoms was performed based on the 100 ns trajectory (Fig 10F). Most residues exhibited low RMSF values (generally below 1.2 Å), indicating overall structural stability of the protein during the simulation. In addition, protein-ligand interaction analysis was conducted to characterize the persistence and nature of intermolecular contacts throughout the simulation (Fig 10G). Several residues, including LYS18, TYR134, LYS179, and ARG183, exhibited high interaction fractions, predominantly mediated by hydrogen bonds, ionic interactions, and water bridges. Time-resolved interaction maps demonstrated that these contacts were maintained across most of the simulation period, supporting the formation of a stable and persistent binding interface. The 2D interaction diagram highlights critical, persistent intermolecular contacts, demonstrating that these interactions effectively anchor ILA within the HMOX1 binding pocket over the entire 100 ns simulation (Fig 10H).

## 4. Discussion

Despite improvements in survival among patients with CRC in recent years, a subset of patients still faces major clinical challenges, including chemotherapy resistance, limited benefit from immunotherapy, and postoperative recurrence [36]. Accordingly, identifying new molecular targets and developing effective intervention concepts remain important priorities in CRC research. Growing evidence supports the involvement of the gut microbiota and its metabolites in CRC initiation and progression. Microbial metabolites act as key mediators of microbiota-host interactions and have therefore attracted increasing interest as potential contributors to CRC-related biological processes. Tryptophan, for example, can be converted by gut microbes into multiple bioactive indole derivatives, including indole, ILA, indole-3-aldehyde (IAld), and indole-3-propionic acid (IPA), which have been reported to shape the TME [5]. ILA has been reported to strengthen intestinal barrier function and attenuate inflammatory responses through AhR signaling, with potential relevance to CRC-related phenotypes [8]. However, the precise molecular mechanisms linking ILA to CRC-associated regulatory processes remain incompletely understood.

This study represents the first systematic elucidation of the potential mechanism underlying ILA in CRC treatment, utilizing an integrative approach combining network pharmacology, machine learning, molecular docking, and molecular dynamics simulations. These methodologies are particularly well-suited for deciphering complex metabolite-disease interactions, as they enable the comprehensive identification of disease- and drug-related targets while uncovering key signaling pathway networks [11]. In this study, four critical targets (EPHA2, HMOX1, MMP3, and PARP1) were identified (Fig 11). We elucidated the potential CRC-associated regulatory mechanisms of ILA through target modulation and discussed its possible immunological relevance, offering new mechanistic perspectives on gut microbiota-derived metabolites in CRC.

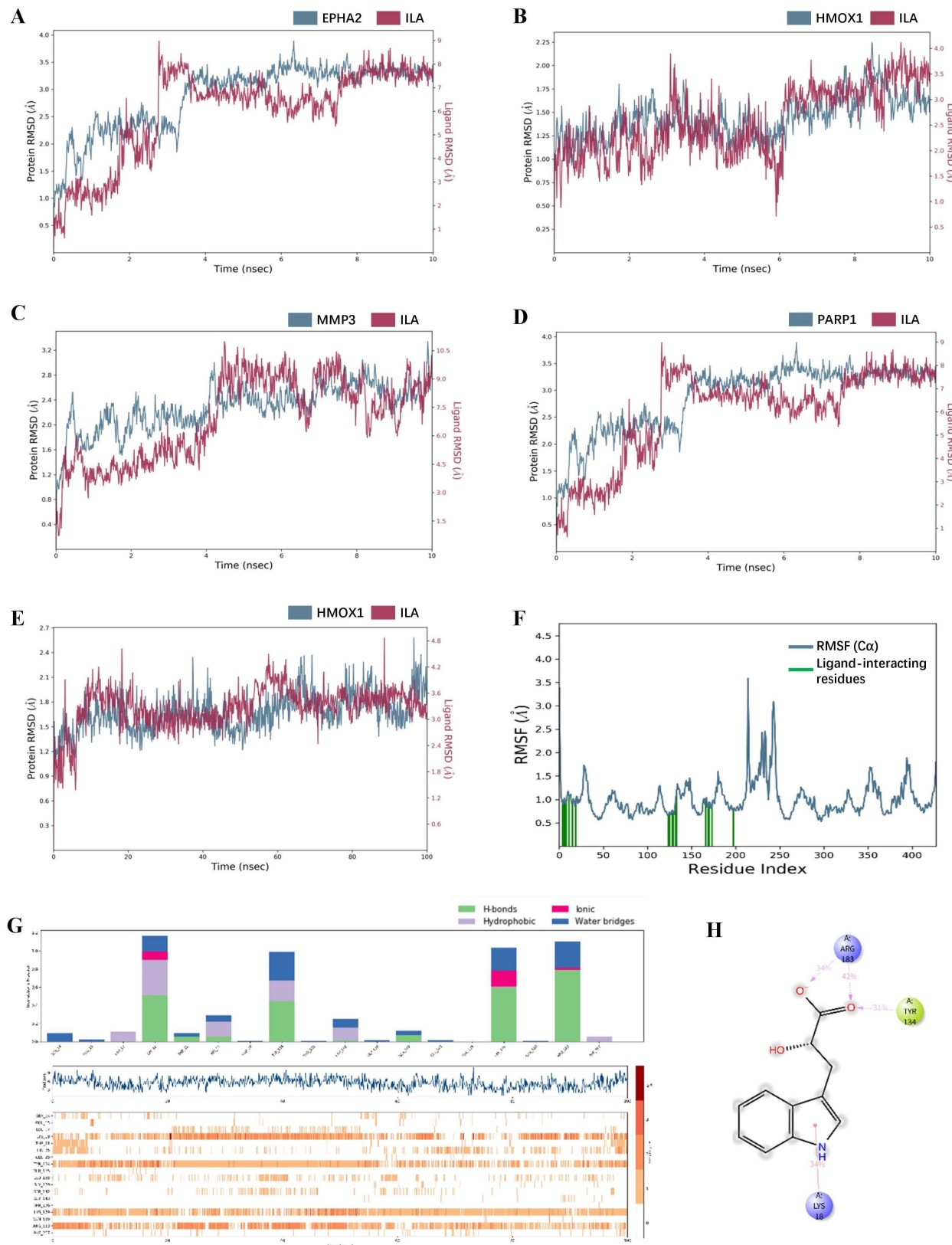

**Fig 10. Molecular dynamics simulations of ILA with hub genes. (A-D)** Comparative RMSD of Cα Atoms for the Four Protein-ILA Complexes over a 10 ns Simulation. **(E)** RMSD plots for the Cα atoms of HMOX1 and the heavy atoms of the ligand ILA during a 100 ns simulation, confirming the dynamic

stability of the complex. **(F)** RMSF analysis of the ILA-HMOX1 complex. The blue curve illustrates the fluctuation magnitude of individual amino acid residues; The green vertical bars highlight the specific residues involved in the interaction with ILA. **(G)** Protein-ligand interaction analysis of the ILA-HMOX1 complex. Interaction fraction of key residues with ILA, categorized by interaction type, including hydrogen bonds, hydrophobic contacts, ionic interactions, and water bridges. **(H)** The 2D schematic diagram illustrating the key persistent intermolecular interactions, including π-cation, salt bridges, and hydrogen bonds, that anchor ILA within the HMOX1 binding pocket throughout the 100 ns trajectory.

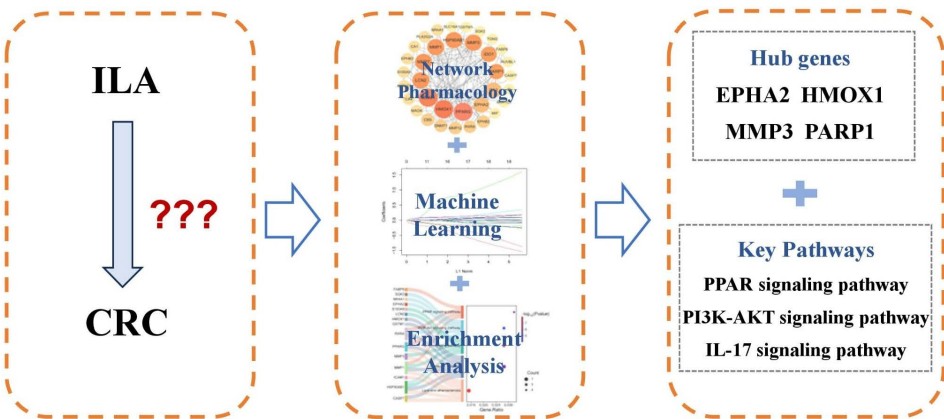

**Fig 11. Potential hub genes and pathways associated with ILA in CRC.**

### 4.1. Potential CRC-associated mechanisms linked to ILA via multi-target regulation

**4.1.1. Oxidative stress regulation.** Heme oxygenase 1 (HMOX1) is an antioxidant enzyme that catalyzes heme degradation into carbon monoxide, biliverdin, and ferrous iron. Increasing evidence links HMOX1 to ferroptosis, a regulated form of cell death driven by iron-dependent lipid peroxidation [37]. By shaping oxidative stress, inflammatory signaling, and apoptosis, the HMOX1 protein has been implicated in tumor cell survival, proliferation, and metastatic behavior. For example, Shenqi Sanjie Granules were reported to induce ferroptosis in colon cancer cells through HMOX1 upregulation, with associated suppression of tumor growth and metastasis [38]. Recent in vitro and in vivo studies suggest that ILA has antioxidant and anti-inflammatory properties. Mechanistic work indicates that ILA can activate the Nrf2 pathway and increase expression of antioxidant defense genes in intestinal epithelial cells [39]. Additional studies suggest that ILA activates Nrf2 via an AhR-dependent mechanism, increases HMOX1 expression in HT-29 colon cancer cells, and enhances tight-junction protein expression, thereby ameliorating LPS-induced intestinal barrier injury [40]. Together, these reports support the plausibility that ILA-associated signaling could be linked to HMOX1-related oxidative stress programs in CRC-associated contexts.

**4.1.2. DNA damage repair regulation.** Poly (ADP-ribose) polymerase 1 (PARP1) participates in DNA repair, transcriptional regulation, and apoptosis. It catalyzes the transfer of ADP-ribose units from $NAD^+$ to target proteins, forming poly(ADP-ribose) (PAR) chains [41]. PARP1 overexpression has been reported in multiple cancer types and has been associated with tumor progression, metastasis, and angiogenesis [42]. Clinically, PARP inhibitors can impair DNA repair and promote apoptosis, either as monotherapy or in combination regimens [43]. Lin et al. reported that a PARP inhibitor reduced metastatic nodules in several organs in CRC nude mouse models, supporting PARP1 as a therapeutically relevant target in CRC [44]. A recent study reported that the indole ring of tryptophan can form stable π-π stacking and hydrogen-bond interactions within the PARP1 active site, potentially mimicking aspects of $NAD^+$ binding and reducing PARP1 activity, which may contribute to DNA repair defects [45]. In our dataset, PARP1 expression was elevated

in CRC samples, and molecular docking suggested that ILA can adopt a plausible binding pose with PARP1 through hydrogen bonding. These results provide a structural rationale for a potential association between ILA-PARP1 interactions and CRC-relevant DNA damage response pathways.

### 4.1.3. Extracellular matrix remodeling.

Ephrin type-A receptor 2 (EPHA2) and matrix metalloproteinase 3 (MMP3) may be jointly related to extracellular matrix (ECM) remodeling and TME reprogramming during CRC progression. EPHA2 is frequently overexpressed in malignancies including breast, prostate, and colorectal cancer, and higher expression is often associated with invasiveness and poor prognosis [46]. EPHA2 has been implicated in metastasis through mechanisms involving EMT and ECM remodeling. In CRC, EPHA2 overexpression has been associated with liver metastasis and adverse clinical outcomes [47]. Cholic acid-tryptophan conjugates have been reported as EPHA2 antagonists [48]. However, direct evidence for ILA-EPHA2 interactions remains limited. In our analysis, EPHA2 was overexpressed in CRC samples, and docking suggested a plausible ILA-EPHA2 interaction. These computational observations raise the possibility that ILA could be linked to EPHA2-associated ECM programs,

MMP3 encodes a matrix metalloproteinase involved in ECM degradation and remodeling [49]. Elevated MMP3 expression has been reported in CRC tumor tissues and is associated with tumor progression and poor prognosis [50]. Although ILA-MMP3 interactions have not been well characterized, IAld and IPA-two other tryptophan-derived indole metabolites, were reported to mitigate IL-1β-induced chondrocyte inflammation and ECM degradation, partly via suppression of inflammatory mediators and MMP3 expression [51–52]. Given the structural similarity among indole metabolites, it is reasonable to hypothesize that ILA could be linked to MMP3-related ECM remodeling pathways in CRC, although direct validation is required.

Across the integrative machine learning and PPI analyses, EPHA2, HMOX1, MMP3, and PARP1 were consistently prioritized as hub genes connecting ILA-associated targets with CRC-related pathological processes. Although direct experimental evidence for ILA binding to these proteins in CRC is currently lacking, convergence across network topology, expression profiling, functional enrichment, immune infiltration analysis, and structural simulations suggests that these genes represent biologically meaningful candidates rather than isolated computational signals. Functionally, EPHA2 and MMP3 are associated with oncogenic signaling and ECM remodeling, PARP1 is central to DNA damage responses and replication stress, and HMOX1 is involved in oxidative stress regulation and inflammatory homeostasis. These findings suggest that ILA may be involved in CRC-associated regulatory processes by coordinately modulating oxidative stress responses, DNA damage repair, and extracellular matrix dynamics, thereby providing mechanistic insight into the potential role of gut microbiota-derived metabolites as adjuvant regulatory strategies in CRC.

## 4.2. Predicted modulation of CRC-associated signaling pathways by ILA

In this study, KEGG and GO enrichment analyses were performed on the 39 ILA-CRC common targets. The enriched terms were mainly related to the PPAR signaling pathway, PI3K–AKT signaling pathway, and IL-17 signaling pathway. In addition, the four hub genes showed associations with DNA replication and fatty acid metabolism signatures in the transcriptomic analyses.

ILA has been reported to exhibit antioxidant and anti-inflammatory properties in experimental systems, and these effects may involve multi-target pathway modulation that contributes to intestinal homeostasis [53]. The PPAR pathway plays a role in lipid metabolism and inflammatory signaling, and dysregulation of this pathway has been linked to CRC development. Lian et al. reported that ILA reduced lipid peroxidation products and alleviated cardiac toxicity in mice [54]. The PI3K-AKT pathway is a major signaling axis controlling proliferation, metabolism, and migration. Prior work suggests that ILA can activate PI3K-AKT signaling via AhR in some contexts, which may be related to inflammatory regulation and macrophage phenotypes [10]. IL-17 is a proinflammatory cytokine produced by Th17 cells and has been implicated in tumor-promoting inflammation and immune evasion [55]. Inflammatory bowel disease (IBD) is a recognized risk factor for colitis-associated CRC [56]. Mechanistic studies suggest that ILA can reduce LPS/TNF-α-induced IL-8 production in

intestinal epithelial cells, which may be relevant to inflammation-driven carcinogenesis [39]. Other studies reported that ILA can target RORγt, inhibit Th17 differentiation, and reduce IL-17 signaling activation in CRC mouse models [57]. Wang and colleagues reported that ILA supports intestinal barrier integrity through coordinated activation of AhR, increased tight-junction protein expression, and inhibition of NF-κB activity, together with reduced proinflammatory cytokines and increased IL-10 in IBD-related settings [38,58]. These findings suggest that ILA may have potential mechanistic implications for the prevention and treatment of CRC through the aforementioned signaling pathways.

### 4.3. Immune-related findings and immune infiltration analysis

Accumulating evidence indicates that immune cell infiltration is associated with prognosis and treatment response in CRC [59]. Alterations in ECM composition and immune cell distribution in the TME can contribute to immunosuppressive states and tumor progression [60,61]. In our study, immune infiltration analysis identified statistically significant correlations between hub gene expression and inferred immune cell proportions. EPHA2 and MMP3 expression levels were positively correlated with activated mast cells. HMOX1 expression correlated positively with neutrophils and negatively with resting memory $CD4^+T$ cells and regulatory T cells. PARP1 expression correlated negatively with resting dendritic cells and M1 macrophages.

Gut microbiota-derived metabolites have been reported to influence immune-related signaling through receptors such as G protein-coupled receptors and AhR, thereby shaping immune responses in various disease contexts [59]. Dendritic cells are antigen-presenting cells that support $CD8^+T$-cell activation, and prior experiments suggested that ILA can enhance $CD8^+T$-cell-associated responses via IL12A regulation in dendritic cells under specific conditions [62]. Macrophages are also key TME components with diverse phenotypes in tumor initiation and progression [63]. ILA has been identified as an AhR ligand, and AhR signaling has been implicated in macrophage phenotypic balance and inflammatory regulation. Li et al. reported reduced expression of the AhR-responsive gene CYP1B1 in CRC based on GEPIA analysis, while in vitro work suggested that ILA may be involved in macrophage differentiation and inflammatory responses in colitis-associated tumor models [64]. Taken together, while these previous findings provide biological context supporting a potential link between ILA-related pathways and immune regulation, the immune infiltration results in this study should be interpreted cautiously. Direct immunomodulatory effects and causal mechanisms require further validation through targeted in vitro and in vivo experiments.

### 4.4. Research limitations and future directions

Through an integrative framework combining network pharmacology, machine learning, molecular docking, and molecular dynamics analyses, this study systematically explored the potential targets and molecular mechanisms of ILA in CRC. Nevertheless, several limitations should be acknowledged. First, although the four hub genes identified were consistently supported by multiple computational approaches, they were primarily derived from bioinformatic analyses and therefore require further experimental validation. Future studies employing in vitro assays and in vivo models will be necessary to confirm the functional roles of these targets and to elucidate their causal involvement in CRC-associated regulatory processes linked to ILA. Second, transcriptomic data provide gene-level insights, but additional multi-omics layers, including proteomics and metabolomics, may offer a more comprehensive view of ILA-related regulatory networks. Integrating these layers may help clarify whether predicted target-pathway relationships are reflected at the protein and metabolite levels.

## 5. Conclusion

By integrating network pharmacology, machine learning, molecular docking, and dynamics simulations, this study systematically elucidates the predicted mechanistic role of the gut microbiota-derived metabolite ILA in CRC. Four hub genes were identified as key molecular nodes potentially associated with CRC-related regulatory processes involving ILA. These findings provide mechanistic insight into how ILA may participate in CRC-associated molecular regulation through coordinated, multi-target interactions. Importantly, this study not only advances the understanding of the molecular mechanisms

underlying ILA activity in CRC, but also highlights the broader potential of gut microbiota-derived metabolites as promising candidates for adjuvant regulatory strategies in CRC.

## Supporting information

**S1 Data. S1 File. The summary of targets of ILA.** S2 File. The summary of targets of CRC. S3 File. KEGG and GO analysis. S4 File. Feature importance rankings of hub genes. S1 Fig. The PCA plots before and after batch correction. S2 Fig. KEGG and GO analysis of 39 common targets.
(ZIP)

## Author contributions

**Conceptualization:** jie li.

**Data curation:** Jian Zhang, Zhijian Ren.

**Formal analysis:** Jun Ke.

**Investigation:** Jian Zhang, Zhijian Ren.

**Methodology:** Jian Zhang.

**Project administration:** Cuncheng Feng.

**Software:** Jun Ke, Zhijian Ren.

**Supervision:** Cuncheng Feng.

**Visualization:** jie li, Jun Ke.

**Writing – original draft:** jie li.

**Writing – review & editing:** jie li, Jian Zhang, Cuncheng Feng.

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
