## [Decision Letter · Decision Letter 0]

24 Dec 2025

Potential mechanism prediction of indole-3-lactic acid against colorectal cancer based on network pharmacology, machine learning and molecular docking

PLOS One

Dear Dr. Feng,

Thank you for submitting your manuscript to PLOS ONE. After careful consideration, we feel that it has merit but does not fully meet PLOS ONE’s publication criteria as it currently stands. Therefore, we invite you to submit a revised version of the manuscript that addresses the points raised during the review process.

This study needs Major revision, after reviewing the manuscript and comments from the reviewers. Both reviewers find the overall study technically sound and the manuscript generally well organized; however, substantial revisions are required to ensure that conclusions are appropriately framed for an in silico study and that the Methods contain sufficient detail for full assessment and reproducibility. The manuscript also requires careful language editing to correct grammatical issues and improve clarity.

We look forward to receiving your revised manuscript.

Kind regards,

Kayode Raheem

Guest Editor

PLOS One

Journal Requirements:

2.Please note that PLOS One has specific guidelines on code sharing for submissions in which author-generated code underpins the findings in the manuscript. In these cases, we expect all author-generated code to be made available without restrictions upon publication of the work. Please review our guidelines at https://journals.plos.org/plosone/s/materials-and-software-sharing#loc-sharing-code and ensure that your code is shared in a way that follows best practice and facilitates reproducibility and reuse.

5. We note that Figure 1 in your submission contain copyrighted images. All PLOS content is published under the Creative Commons Attribution License (CC BY 4.0), which means that the manuscript, images, and Supporting Information files will be freely available online, and any third party is permitted to access, download, copy, distribute, and use these materials in any way, even commercially, with proper attribution. For more information, see our copyright guidelines: http://journals.plos.org/plosone/s/licenses-and-copyright.

Please upload the completed Content Permission Form or other proof of granted permissions as an 'Other' file with your submission.

Additional Editor Comments:

Dear Dr. Feng,

Thank you for submitting your manuscript, “Potential mechanism prediction of indole-3-lactic acid against colorectal cancer based on network pharmacology, machine learning and molecular docking” (PONE-D-25-56178), to PLOS ONE.

We invite you to revise the paper, carefully addressing the comments from the reviewers and the editor. After considering the reports, my decision is Major revision. The overall computational workflow is of interest and the manuscript is generally well structured, but substantial revisions are needed to ensure that conclusions are appropriately framed for an in silico study and that the Methods provide sufficient detail for full assessment and reproducibility. When this revision is ready, please submit the updated manuscript and a point-by-point response. This will allow the reviewers and editor to evaluate how each concern has been addressed and, where appropriate, to determine whether additional external review is required. The reviewers' comment is attached

Reviewers' comments:

Reviewer's Responses to Questions

**Comments to the Author**

1. Is the manuscript technically sound, and do the data support the conclusions?

Reviewer #1: Yes

Reviewer #2: Yes

2. Has the statistical analysis been performed appropriately and rigorously?

Reviewer #1: Yes

Reviewer #2: Yes

3. Have the authors made all data underlying the findings in their manuscript fully available?

Reviewer #1: Yes

Reviewer #2: No

4. Is the manuscript presented in an intelligible fashion and written in standard English?

Reviewer #1: Yes

Reviewer #2: Yes

Reviewer #1: From introduction to conclusion, the manuscript is well-structured and clear. A gut microbiota-derived metabolite, Indole-3-lactic acid (ILA), may treat colorectal cancer (CRC). The authors present a complete history. The study's goals, methods, and results are organized. A more extensive description of several portions would improve clarity before acceptance.

Title

• The title is written in a passive voice, which can make it sound less engaging and less clear. Consider rephrasing the title.

• The title is a bit long and should be rephrased to be more concise.

• Consider rephrasing the title to focus more on the research question or the main findings.

Abstract:

• Rephrase the sentence "Our study systematically elucidated..." to be more concise and impactful.

• Consider adding a brief summary of the study's main findings.

Introduction:

• Add more specific details about the current state of CRC research and how ILA fits into this landscape.

• Reorganize the paragraph discussing ILA's biological activities and mechanisms to improve flow.

• Add transitional phrases or sentences to connect the ideas between paragraphs.

• Rephrase the sentence "In this study, we integrated..." to be more concise and focused on the main objective.

Materials and Methods:

• Provide more detail on the databases and tools used, including their versions and specific parameters.

• Clarify the criteria for selecting the soft threshold, β.

• Provide more information on the PPI network analysis, including visualization parameters.

• Justify the use of specific machine learning algorithms.

• Provide more detail on the molecular docking analysis, including specific parameters used, such as the grid size and ligand flexibility.

• The section mentions that MD simulations were performed using Desmond in the Schrödinger Suite 2021-4. However, it would be helpful to provide more information on the specific protocol used, such as the simulation time, temperature, and pressure. I invite authors to clarify the MD simulation protocol.

The authors should revise the Methods section to include all necessary details regarding parameter selection, statistical thresholds, and multiple testing corrections to allow for full assessment and reproducibility.

Results

• Some figures and tables are not clearly described or referenced in the text. For example, Figure S1 is mentioned in the text, but its content is not clearly described. Clarify the presentation of results, including figures and tables.

• The KEGG and GO enrichment analyses are performed on the 39 therapeutic targets. However, it would be helpful to provide more context on why these specific targets were chosen and how they relate to the overall study.

• The machine learning algorithms identify four hub genes (EPHA2, HMOX1, MMP3, and PARP1). However, it would be helpful to discuss the implications of these findings and how they relate to the overall study. I recommend authors to discuss the implications of the machine learning results

• The molecular docking and MD simulations are performed on the four hub genes. Nevertheless, it would be supportive to provide more detail on the specific parameters used and the results obtained.

• The study has several limitations, including the reliance on bioinformatic analyses and the need for further functional validation.

• Clarify the implications of the study's findings and how they relate to the overall research question.

• The text employs various terms to denote the same concept, including "ILA-associated targets" and "ILA-CRC shared targets." Consistent terminology throughout the text would be beneficial.

• The text indicates the utilization of four GEO datasets; however, it lacks clarity regarding the rationale for selecting these particular datasets and the processing methods employed.

• The text states that WGCNA was employed to identify gene modules associated with CRC; however, it lacks clarity regarding the analysis methodology and the parameters utilized.

• The study is significantly dependent on bioinformatic analyses, which may introduce biases and limitations. It is essential to address these potential biases and limitations more explicitly.

• Phrases like "Collectively", "Notably", “In conclusion”, and "Furthermore" are used frequently throughout the manuscript, which can make it sound like it was written by an AI model.

Reviewer #2: Manuscript Title

Potential mechanism prediction of indole-3-lactic acid against colorectal cancer based on network pharmacology, machine learning and molecular docking

Overall Assessment

This manuscript presents a comprehensive in silico investigation of the potential mechanisms by which the gut microbiota–derived metabolite indole-3-lactic acid (ILA) may influence colorectal cancer (CRC). The authors integrate network pharmacology, transcriptomic analysis, machine learning–based feature selection, molecular docking, and molecular dynamics (MD) simulations. Given the increasing interest in gut microbiota–derived metabolites and their roles in cancer biology, this study provides a computational framework to identify potential molecular targets and pathways associated with ILA. Thus, it offers a useful hypothesis-generating resource for future experimental studies. However, several major issues related to interpretation, methodological transparency, and overstatement of conclusions should be addressed before the manuscript can be considered suitable for publication.

Major Comments

1. Overinterpretation of Computational Findings

Throughout the Abstract, Results, Discussion, and Conclusion, the manuscript frequently implies confirmed “therapeutic effects” of ILA against CRC e.g.

• “Our study systematically elucidated the potential therapeutic effect of ILA on CRC…” (Abstract)

• “ILA may exert its anti-CRC effects by targeting these proteins” (Results 3.8)

Given that the study is entirely computational, these statements overstate the strength of the evidence.

Recommendation:

The authors should revise the language throughout the manuscript to clearly frame the findings as predictive, hypothesis-generating, or putative mechanisms. Replace “therapeutic effect” with “predicted mechanism”, “putative targets”, or “hypothesized regulatory roles”. Claims of therapeutic efficacy should be avoided unless supported by experimental validation.

2. Target Prediction Strategy Requires Additional Justification

In Section 2.1 (Collection of ILA targets), predicted targets from multiple databases are combined without reporting confidence scores, prediction probabilities, or overlap frequency across databases.

Recommendation:

The authors should provide additional information on target confidence, such as:

• The number of databases supporting each target

• Probability or score thresholds (where available)

• A rationale for including low-confidence predictions

3. Machine Learning Methodology Lacks Transparency

In Section 2.7, the use of LASSO, Random Forest, and SVM-RFE is appropriate; however, important methodological details are missing:

• Handling of class imbalance between normal and CRC samples is not described.

• Feature scaling or normalization prior to SVM analysis is not reported.

• The Random Forest importance threshold (>0.3) is not justified.

Recommendation:

The authors should clarify preprocessing steps, provide justification for parameter choices, and include additional information to ensure reproducibility. Reporting cross-validation strategies and performance metrics in greater detail would strengthen this section.

4. Interpretation of Immune Infiltration Analysis

The immune infiltration analysis (Section 3.7) identifies correlations between hub gene expression and immune cell proportions. However, the discussion occasionally implies mechanistic regulation of immune cell behavior by ILA or hub genes.

Recommendation:

The authors should explicitly state that these findings are correlational. Causal language should be avoided, and conclusions regarding immune modulation should be clearly framed as speculative since these claims was not substantially validated in an invitro or in vivo model.

5. Molecular Docking and Molecular Dynamics Analysis

The docking and MD simulations support binding plausibility between ILA and the identified hub proteins. However, the rationale for extending the MD simulation to 100 ns for only the ILA–HMOX1 complex, while limiting others to 10 ns, is not sufficiently explained.

Recommendation:

The authors should justify the selection of HMOX1 for extended simulation and discuss the limitations of short MD trajectories for the other complexes. Additional stability metrics or binding free energy calculations would further strengthen this section. Report RMSF, hydrogen bond occupancy, and binding free energy (MM-PBSA). Also Include control ligands or known inhibitors of CRC for benchmarking.

Minor Comments

1. Data Availability Statement

The statement “All data are available on request from the authors” does not align with PLOS ONE data-sharing recommendations. Public access to analysis scripts (e.g., GitHub, Zenodo) or processed data should be provided where possible.

2. Redundancy in the Discussion

Several mechanistic explanations (e.g., AhR/Nrf2 signaling) are repeated across multiple subsections. The Discussion could be streamlined to improve clarity.

3. Language and Formatting

Minor grammatical errors and overly long sentences are present, particularly in the Discussion. Gene and protein nomenclature should be consistently formatted.

**Do you want your identity to be public for this peer review?** For information about this choice, including consent withdrawal, please see our Privacy Policy

Reviewer #1: No

Reviewer #2: **Yes:** Modinat Aina Abayomi

---

## [Author Response · Author response to Decision Letter 1]

22 Jan 2026

Dear Editors:

Thank you for giving us the opportunity to submit a revised draft of our manuscript “Potential mechanism prediction of indole-3-lactic acid against colorectal cancer based on network pharmacology, machine learning and molecular docking” (Submission ID PONE-D-25-56178) for publication in PLOS One. We appreciate the time and effort that you and the reviewers dedicated to providing feedback on our manuscript and are grateful for the insightful comments and valuable improvements to our paper. We have read your comments carefully and have made some modifications which hopefully could meet your expectations. The revised portions are highlighted in red in the manuscript. The main modifications in the paper are summarized and the response to the reviewers’ comments are as follows:

Reviewer 1, Point 1

1.Comment: (Title: The title is written in a passive voice, which can make it sound less engaging and less clear. Consider rephrasing the title. The title is a bit long and should be rephrased to be more concise. Consider rephrasing the title to focus more on the research question or the main findings.)

Response: We sincerely appreciate your thorough review and valuable suggestions regarding our manuscript. Following your advice, we have rephrased the title to be more concise, active, and focused on the core research findings. We have revised the original title to: “Integrative network pharmacology and machine learning identify potential targets of indole-3-lactic acid in colorectal cancer”. We have updated the title on the Title Page and throughout the manuscript submission system.

Reviewer 1, Point 2

2.Comment: (Abstract: Rephrase the sentence "Our study systematically elucidated..." to be more concise and impactful. Consider adding a brief summary of the study's main findings.)

Response: We sincerely thank your for this insightful suggestion regarding the Abstract. Following your suggestion, we have carefully revised the Abstract and rewritten the concluding sentence to be more concise and impactful. The revised sentence reads as follows: “In conclusion, these results highlight EPHA2, HMOX1, MMP3, and PARP1 as candidate targets and suggest that ILA may influence CRC-related signaling, metabolic programs, and immune contexture, providing a theoretical foundation for developing gut microbiota-derived metabolites as novel anticancer strategies”. We have updated the Abstract in the revised manuscript.

Reviewer 1, Point 3

3.Comment: (Introduction: Add more specific details about the current state of CRC research and how ILA fits into this landscape.)

Response: We sincerely thank your for this insightful suggestion regarding the Introduction. Following your suggestion, we have rewritten the Introduction, with the main revisions focusing on adding and strengthening the following two aspects: (1) Summarize the current therapeutic and research challenges in CRC, including molecular heterogeneity and treatment resistance. (2) Better position gut microbiota–derived metabolites as an emerging research direction relevant to tumor metabolism and the immune microenvironment. We further clarified how ILA, as a tryptophan-derived microbial metabolite with reported anti-inflammatory, antioxidant and immunomodulatory activities, conceptually fits this framework and why a systematic target- and pathway-level investigation is needed. These revisions have been incorporated in the Introduction section. Please see the specific modifications in the revised manuscript.

Reviewer 1, Point 4

4.Comment: (Materials and Methods: Provide more detail on the databases and tools used, including their versions and specific parameters.)

Response: Thank you very much for pointing this out, and we think this is an excellent suggestion. In accordance with your suggestion, we have thoroughly updated the "Materials and Methods" section to ensure full transparency and reproducibility. The specific revisions are as follows: (1) Database access: We have added the specific URLs and last access dates for all online databases. (2) Software versions: Beyond the main R software version (v4.4.2), we have now specified the exact version numbers for all critical R packages used in the analysis. (3) Parameter specification: We have elaborated on the specific parameters for the computational models used in this study. Specifically, we provided the detailed workflow for the PPI network analysis (Section 2.5); clarified the rationale for the selected machine learning algorithms alongside their hyperparameter tuning settings (Section 2.7); and supplemented detailed information regarding the molecular docking and molecular dynamics simulations (Sections 2.10 and 2.11). These modifications have significantly improved the clarity and readability of our manuscript. Please see the specific modifications in the revised manuscript.

Reviewer 1, Point 5

5.Comment: (Materials and Methods: Clarify the criteria for selecting the soft threshold, β.)

Response: Thank you for your pointing this out. We apologize for not making it clear. We have added a detailed description in Section 2.4 of the criteria used to select the soft-thresholding power (β) in the WGCNA analysis, as follows: “We determined the optimal soft threshold power (β) using the pickSoftThreshold function in the WGCNA R package. The criterion for selection was to achieve a scale-free topology fit index (R2) of at least 0.85 while maintaining a reasonable mean connectivity. Consequently, a power of β=X was selected based on the scale independence and mean connectivity plots. ”

Reviewer 1, Point 6

6.Comment: (Materials and Methods: Provide more information on the PPI network analysis, including visualization parameters.)

Response: Thank you for your pointing this out. We apologize for not making it clear. We have added a detailed description in Section 2.5 of the parameters and visualization workflow used for the PPI analysis, as follows: “The PPI data were downloaded from the STRING database in TSV format and imported into Cytoscape (version 3.9.1) for network construction and visualization. Network topological properties were analyzed using the ‘Analyze Network’ tool with the network treated as undirected (‘Treat network as undirected’). Node centrality measures were then calculated using the CytoNCA plugin (version 2.1.6), with degree centrality used as the primary metric. Isolated nodes without interactions (degree = 0) were excluded from the network. Finally, the ‘Style’ panel was used to map node attributes for visualization: node size and color were scaled according to degree centrality values, with larger and darker nodes indicating higher connectivity.”

Reviewer 1, Point 7

7.Comment: (Materials and Methods: Justify the use of specific machine learning algorithms.)

Response: Thank you very much for your comments. We have added a detailed justification in Section 2.7 for using the three machine-learning algorithms (LASSO, RF, and SVM-RFE), as follows: “Specifically, LASSO employs L1 regularization to mitigate multicollinearity and induce sparsity in high-dimensional datasets; RF was selected to capture complex non-linear interactions; and SVM-RFE identifies an optimal feature subset to maximize classification performance. These three algorithms are complementary, and their combined use reduces the bias inherent to any single model, thereby improving the robustness of the screening results.”

Reviewer 1, Point 8

8.Comment: (Materials and Methods: Provide more detail on the molecular docking analysis, including specific parameters used, such as the grid size and ligand flexibility.)

Response: We sincerely appreciate the valuable comments. In response to your suggestion, we have updated Section 2.10 with the following detailed parameters: (1) Grid generation: The receptor grids were generated using the Receptor Grid Generation panel in Glide. The center of the grid box was defined by the centroid of the co-crystallized ligand within the active site of each target protein. The bounding box was sized to sufficiently enclose the active site and accommodate the ligand. The specific grid dimensions for each target are provided in Table 2. (2) Ligand flexibility: The Glide Standard Precision mode was employed, which internally generates multiple conformations for the ligand. The sampling included the exploration of ring conformations and nitrogen inversions. To account for protein flexibility implicitly and allow for minor steric clashes, the van der Waals radii of the ligand atoms were scaled by a factor of 0.80 with a partial charge cutoff of 0.15, while the receptor atoms were scaled by 1.0. (3) The resulting poses were subjected to post-docking minimization, and the best-scored pose (lowest GlideScore) was selected for further analysis. These revisions have been incorporated into the manuscript Section 2.10.

Reviewer 1, Point 9

9.Comment: (Materials and Methods: The section mentions that MD simulations were performed using Desmond in the Schrödinger Suite 2021-4. However, it would be helpful to provide more information on the specific protocol used, such as the simulation time, temperature, and pressure. I invite authors to clarify the MD simulation protocol.)

Response: Thank you very much for pointing this out, and we agree that a clearer and more detailed description of the MD protocol is essential for reproducibility. We have therefore revised Section 2.11 to explicitly report the key simulation settings, including the simulation time for each system, the ensemble, temperature, and pressure, as well as additional protocol details (system solvation, equilibration procedure, time step and constraints, thermostat, and trajectory recording frequency). Briefly, all simulations were performed in Desmond (Schrödinger Suite 2021-4) using the OPLS4 force field, in a TIP3P water box with 0.15 M NaCl. After minimization and equilibration, four independent 10 ns comparative simulations were conducted for the four protein-ligand complexes, and a 100 ns production run was performed for the ILA-HMOX1 complex under the NPT ensemble at 300 K and 1 atm. These details have been added to the revised manuscript Section 2.11.

Reviewer 1, Point 10

10.Comment: (Materials and Methods: The authors should revise the Methods section to include all necessary details regarding parameter selection, statistical thresholds, and multiple testing corrections to allow for full assessment and reproducibility.)

Response: We sincerely appreciate the valuable comments. Accordingly, we have comprehensively revised the Materials and Methods section to explicitly describe the criteria and parameter settings used at each analytical step. We have incorporated these revisions into Section 2 of the revised manuscript.

Reviewer 1, Point 11

11.Comment: (Results: Some figures and tables are not clearly described or referenced in the text. For example, Figure S1 is mentioned in the text, but its content is not clearly described. Clarify the presentation of results, including figures and tables.)

Response: Thank you for your pointing this out. We apologize for not making it clear. In accordance with your suggestion, we have carefully re-examined the entire Results section. (1) Regarding Figure S1: We have added a detailed description in the main text to explain the changes in PCA plots before and after batch effect correction, demonstrating the reliability of our data integration. (2) General revision: We have systematically checked the citations and descriptions for all figures (Figures 1-11; Figure S1-S2) and tables (Tables 1-2). We revised the manuscript to ensure that every reference to a figure or table is accompanied by a clear interpretation of the specific data trends and their biological significance. Please see the specific modifications in the revised manuscript.

Reviewer 1, Point 12

12.Comment: (The KEGG and GO enrichment analyses are performed on the 39 therapeutic targets. However, it would be helpful to provide more context on why these specific targets were chosen and how they relate to the overall study.)

Response: We sincerely appreciate the valuable comments.

(1) The 39 targets used for GO and KEGG enrichment analyses were not arbitrarily selected but were identified through a stepwise integrative strategy designed to link ILA with CRC in a biologically meaningful manner. Specifically, these 39 targets represent the intersection between 1) putative ILA-related targets predicted using multiple network pharmacology databases and 2) CRC key genes defined by integrating disease-related databases, differential expression analysis, and WGCNA. Therefore, these targets simultaneously reflect the potential molecular actions of ILA and the core pathological features of CRC.

(2) Performing functional enrichment analyses on this refined target set allowed us to focus on biological processes and signaling pathways that are most likely involved in the therapeutic effects of ILA against CRC, rather than generating nonspecific results from broader gene sets. Importantly, the enriched GO terms and KEGG pathways provide mechanistic context for the subsequent identification of hub genes, immune infiltration analysis, and molecular docking.

(3) To clarify this rationale, we have revised the Results section (Section 3.3) and added explanatory text describing the selection criteria and biological significance of the 39 therapeutic targets, as well as their role in the overall study framework.

Reviewer 1, Point 13

13.Comment: (The machine learning algorithms identify four hub genes (EPHA2, HMOX1, MMP3, and PARP1). However, it would be helpful to discuss the implications of these findings and how they relate to the overall study. I recommend authors to discuss the implications of the machine learning results.)

Response: Thank you very much for your comments. In this study, machine learning algorithms were applied not merely as a feature-selection tool but as an integrative strategy to identify the most robust and biologically relevant hub genes among the potential therapeutic targets of ILA against CRC. By combining LASSO regression, RF, and SVM-RFE, we aimed to reduce model-specific bias and improve the reliability of hub gene selection.

(1) The four hub genes identified were consistently supported by multiple independent analyses, including PPI network topology, differential expression and diagnostic performance, gene set enrichment analysis, immune infiltration correlations, and molecular docking and molecular dynamics simulations. This convergence of evidence indicates that these genes represent key molecular nodes linking ILA-associated targets to CRC-related pathological processes, rather than being isolated machine learning outputs.

(2) From a biological perspective, these hub genes are involved in critical processes relevant to CRC progression. Their identification therefore provides mechanistic insight into how ILA may exert anti-CRC effects through coordinated regulation of inflammation, oxidative stress, metabolism, and tumor-host interactions.

(3) We have added a comprehensive discussion at the end of Section 4.1 of the Discussion to improve the interpretability and biological relevance of the machine learning analyses.

Reviewer 1, Point 14

14.Comment: (The molecular docking and MD simulations are performed on the four hub genes. Nevertheless, it would be supportive to provide more detail on the specific parameters used and the results obtained.)

Response: We sincerely appreciate the valuable comments. In accordance with your recommendation, we have made substantial additions to the manuscript in two areas: (1) We have supplemented the text with specific operational parameters for molecular docking and molecular dynamics simulations in the Methods section (Sections 2.10 and 2.11). (2) We have provided a more granular and quantitative description of the simulation outcomes in the Results section (Sections 3.8 and 3.9). We believe these added details significantly enhance the transparency and robustness of our computational findings.

Reviewer 1, Point 15

15.Comment: (The study has several limitations, including the reliance on bioinformatic analyses and the need for further functional validation.)

Response: We sincerely appreciat

---

## [Decision Letter · Decision Letter 1]

23 Feb 2026

Integrative network pharmacology and machine learning identify potential targets of indole-3-lactic acid in colorectal cancer

PONE-D-25-56178R1

Dear Dr. Feng,

I am pleased to inform you that your revised manuscript, **“Integrative network pharmacology and machine learning identify potential targets of indole-3-lactic acid in colorectal cancer****”**  (Manuscript ID: **PONE-D-25-56178R1** ), is **accepted for publication**  in PLOS ONE, contingent upon completion of any outstanding technical requirements.

Kind regards,

Kayode Raheem

Guest Editor

PLOS One

---

## [Editor Report · Acceptance letter]

PONE-D-25-56178R1

PLOS One

Dear Dr. Feng,

I'm pleased to inform you that your manuscript has been deemed suitable for publication in PLOS One. Congratulations! Your manuscript is now being handed over to our production team.

Kind regards,

on behalf of

Dr. Kayode Raheem

Guest Editor

PLOS One